# Maximum Penetration Height and Intrusion Speed of Weak Symmetric Plane Fountains in Linearly Stratified Fluids

Mohammad I. Inam [1], Wenxian Lin [2,*], Steven W. Armfield [3] and Mehdi Khatamifar [2]

[1] Department of Mechanical Engineering, Khulna University of Engineering and Technology, Khulna 9203, Bangladesh; iliasinam@me.kuet.ac.bd

[2] College of Science and Engineering, James Cook University, Townsville, QLD 4811, Australia; mehdi.khatamifar2@jcu.edu.au

[3] School of Aerospace, Mechanical and Mechatronic Engineering, The University of Sydney, Sydney, NSW 2006, Australia

[*] Correspondence: wenxian.lin@jcu.edu.au

**Abstract:** The flow behavior of weak symmetric plane fountains in linearly stratified fluids is studied numerically with three-dimensional simulations over a range of the Froude ($Fr$), Reynolds ($Re$), and stratification numbers ($s$). The two main parameters describing the fountain characterization are the dimensionless maximum fountain penetration height ($z_m$) and intrusion velocity ($u_{int}$), which differ significantly at different flow development stages. It was found that the stratification stabilizes the symmetry of the weak fountains, which makes the fountain become asymmetric at a larger $Fr$ value, and $z_m$ at the fully developed stage continues to increase as a result of the intrusion, which continually changes the ambient fluid stratification features, thus the buoyant force. The evolution of intrusion experiences three distinct stages. Both $Fr$ and $s$ have effects on $z_m$ and $u_{int}$, with the effect of $Fr$ usually larger than that of $s$. The overall impacts of $Fr$ and $s$ can be quantified in terms of $Fr^a s^b$, with $a$ and $b$ varying for different parameters. With numerical results, empirical correlations are produced in terms of $Fr^a s^b$ for each relevant parameter, which generally predict the results very well.

**Keywords:** weak plane fountain; stratification; penetration height; intrusion; symmetry

## 1. Introduction

Fountains are widely encountered in many natural settings and practical applications. Their behavior is also of fundamental significance as our understanding of free shear flows under negatively buoyant force still needs improvement. These make them a topic that has drawn a remarkable research interest. Hunt and Burridge [1] conducted a very comprehensive review on the studies on the behavior of different types of fountains over wide ranges of governing parameters and under different conditions.

A fountain is a flow caused by a fluid injected vertically upward into a large body of fluid which has a smaller density. The buoyancy experienced by the upward fluid opposes its upward velocity resulting in a gradual reduction in velocity up to zero. The height where the upward velocity attains zero is denoted as the maximum fountain penetration height (abbreviated as 'MFPH' hereafter). After the upward fluid attains the MFPH, it reverses the moving direction and descends and falls on the bottom floor to form an intrusion that moves outward along the bottom floor.

The ambient fluid in which a fountain penetrates can be homogeneous or stratified. When the ambient fluid is stratified, the fluids at different heights have different densities. A stable stratification with a constant stratification number is the physical situation in which the fluid densities decrease from the bottom to the top at a constant density gradient. Such a fluid is called a linearly stratified fluid. A fountain is generally characterized by the Reynolds number, $Re$, Froude number, $Fr$, and density stratification parameter, $S_p$, if the

ambient is a linearly stratified fluid (abbreviated as 'LSF' hereafter). These three governing parameters are defined below [2],

$$Re = \frac{W_0 X_0}{\nu},$$ (1)

$$Fr = \frac{W_0}{[gX_0(\rho_0 - \rho_{a,0})/\rho_{a,0}]^{1/2}} = \frac{W_0}{[g\beta X_0(T_{a,0} - T_0)]^{1/2}},$$ (2)

$$S_p = -\frac{1}{\rho_{a,0}}\frac{d\rho_{a,Z}}{dZ}.$$ (3)

$Fr$ in the second expression of Equation (2) is implemented when the Oberbeck–Boussinesq approximation is valid, which is what is assumed in the present study. The temperature stratification parameter, $S$, can also be represented using this approximation, as follows [2],

$$S = \frac{dT_{a,Z}}{dZ} = \frac{S_p}{\beta},$$ (4)

The dimensionless form of $S$ is widely used to generalize the results and defined below [2],

$$s = \frac{d\theta_{a,z}}{dz} = \frac{X_0}{(T_{a,0} - T_0)}S = \frac{X_0}{\beta(T_{a,0} - T_0)}S_p.$$ (5)

Past studies on fountains have focused on round ones in homogeneous fluids (i.e., $S_p = 0$), with the dimensionless MFPH ($z_m$) as the main variable to characterize and quantify the fountain behavior. A round fountain can be 'very weak' when $Fr \lesssim 1$, 'weak' when $1 \lesssim Fr \lesssim 3$, or 'forced' when $Fr \gtrsim 3$ [3,4]. Forced round fountains are found to be significantly different from weak fountains. The readers are referred to the review by Hunt and Burridge [1] for the details.

The behavior of a plane fountain (abbreviated as 'PF' hereafter), which is formed by injecting an upward dense fluid continuously into a lighter ambient fluid, either homogeneous or stratified, from a narrow slot, has also been explored, such as [2,5–9], although it has been apparently far less studied [1,9,10].

Similar to round fountains, a PF in a large container of homogeneous fluid may be either 'very weak' when $Fr \ll 1$, 'weak' when $Fr = O(1)$, or 'forced' when $Fr \gg 1$, as classified by Hunt and Coffey [11].

The studies on PFs have dominantly focused on fountains in homogeneous fluids, although some studies also focused on those in stratified fluids, such as [5,8,12–15]. Research on PFs in stratified fluids has been scarce and mainly focused on those in turbulent regimes, with $z_m$ as the main parameter. In the past several years, we carried out numerical studies on weak and transitional PFs in LSFs to examine their effects on the transition of a PF from symmetric behavior to asymmetric behavior, with $z_m$ as the major parameter characterizing the PF behavior [10,16–18]. Nevertheless, the characterization of weak PFs in stratified fluids with small $Fr$ and $Re$ values, in which only the symmetric behavior is present, is currently rarely understood. Furthermore, the intrusion is an integral part of a weak symmetric PF (abbreviated as 'SPF' hereafter), and its behavior is important due to its significant effects on $z_m$. Nonetheless, intrusion in weak SPF, particularly in stratified ambient fluids, is rarely investigated. The aforementioned points motivate us to investigate the characterization of penetration and intrusion of SPFs in LSFs.

In this work, numerical simulations were executed for weak SPFs in a large body of LSF over a range of $Fr$ and $s$ as a fixed small $Re$ to ensure the fountains were laminar and to study their effects on the MFPH and the intrusion behavior of weak SPFs in LSFs.

## 2. Methodology

The physical and numerical model is assumed to be a rectangular domain with the dimensions $H \times B \times L$, in which an initially quiescent fluid at a constant stratification parameter value is fully filled. At the bottom center of the domain, as the source for the PF, a slot with a half-width of $X_0$ exists. The remainder of the bottom and top surfaces (in the $X$-$Y$ plane, at $Y = H$) are rigid, non-slip and adiabatic. The periodic boundary condition is applied to the two vertical sidewalls (in the $X$-$Z$ plane, at $Y = \pm B/2$), and the two vertical surfaces in the $Y$-$Z$ plane, at $X = \pm L/2$ (front and back boundaries), are considered to be outflows. The origin of the Cartesian coordinate systems is at the bottom center, with gravity in the negative $Z$-direction. At time $t = 0$, to initiate the PF flow, a fluid at the temperature $T_0$, which is smaller than $T_{a,0}$, with a uniform velocity $W_0$ is injected upward from the slot into the domain, which is subsequently continued during the whole time period of a particular numerical simulation run.

The details of the flow governing equations and the assigned initial and boundary conditions are given in the Supplementary Materials, which also contain further details of the methodology, including the discretization of the governing equations using a finite volume method, the solution solvers, the construction of the non-uniform meshes, the mesh and time-step independence testing, and the values of relevant parameters used for the numerical simulations. All numerical simulations were performed using ANSYS Fluent 13.0.

A total of 32 numerical simulation runs have been carried out, all at the fixed $Re = 100$: 6 runs for $s = 0.1$ (with $Fr = 1, 1.5, 2, 2.75, 2.875, 3$); 6 runs for $s = 0.2$ (with $Fr = 1, 2, 2.5, 3, 3.25, 3.5$); 6 runs for $s = 0.3$ (with $Fr = 1, 2, 3, 3.5, 3.75, 4$); 7 runs for $s = 0.4$ (with $Fr = 1, 2, 3, 4, 4.35, 4.5, 5$); and 7 runs for $s = 0.5$ (with $Fr = 1, 2, 3, 4, 4.75, 4.85, 5$).

## 3. Qualitative Observation

### 3.1. Development of Temperature Field

The development of the temperature field in the domain is shown by the evolution of the transient temperature contours of the PF at $Fr = 3.25$ and $s = 0.1$ on three specific planes, as illustrated in Figure 1. It can be observed that for all the instants illustrated, the $X = 0$ surface in the $Y$-$Z$ plane is the symmetry surface (as shown in the first column of Figure 1). From the second and third columns, it can be seen that along the $Y$ direction, the temperature gradient is zero. This observation is unlike the asymmetric behavior reported for PFs in stratified fluids at larger $Fr$ and $Re$ [10]. This finding implies that for the specific fountain presented, the flow and temperature pattern remains symmetric irrespective of the instant selected in contrast to an asymmetric fountain [10,16,17].

### 3.2. Effect of Fr

Figure 2 demonstrates the influence of $Fr$ on the SPFs at the fully developed stage (abbreviated as 'FDS' hereafter). The symmetry observed for different instances in Figure 1 can be seen for different $Fr$ cases. By increasing $Fr$, the momentum flux of the injected fluid becomes larger, which leads to increased $z_m$. A typical feature of a weak fountain is having indistinguishable upflow and the downflow that makes little entrainment of the ambient fluid into the core region of the fountain fluid [2,6]. The first and second columns of Figure 2 also depict the variation of intrusion thickness. The intrusion thickness increases considerably compared to the fountain height, especially for a small $Fr$. This behavior has a major impact on $z_m$ that will be discussed subsequently.

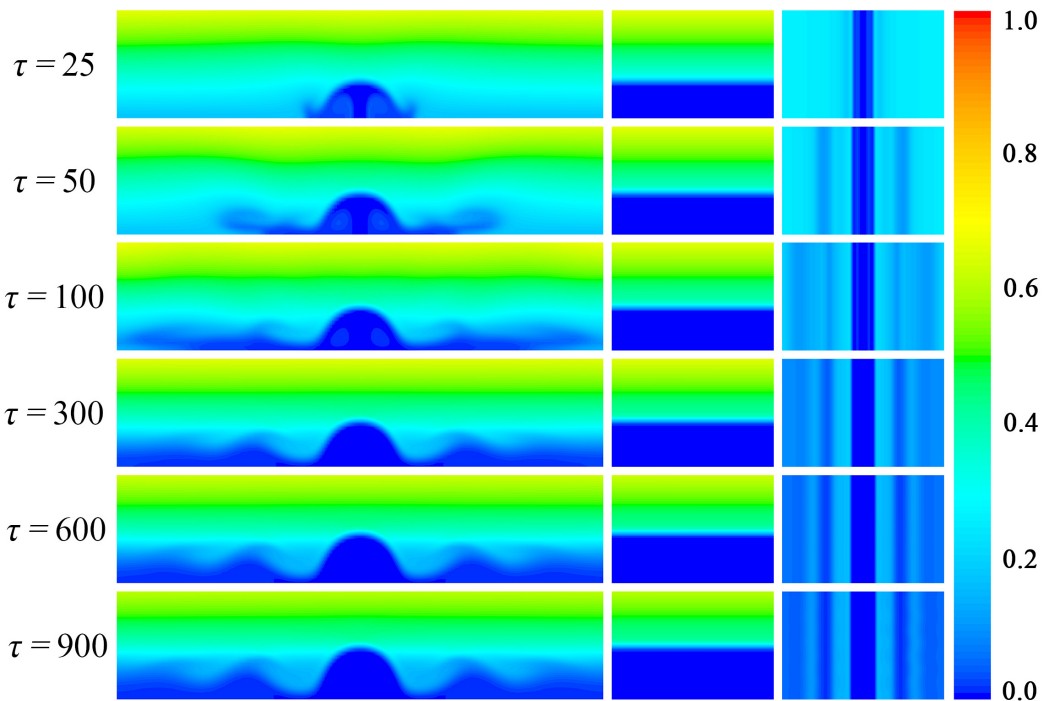

**Figure 1.** Evolution of the transient temperature contours of the PF at *Fr* = 3.25 and *s* = 0.1 at *Y* = 0 in the *X*-*Z* plane (first column), *X* = 0 in the *Y*-*Z* plane (second column), and *z* = 0.5$z_{m,i}$ in the *X*-*Y* plane (third column). Temperatures are non-dimensionalized with $[T(Z) - T_0]/(T_{a,z=60} - T_0)$.

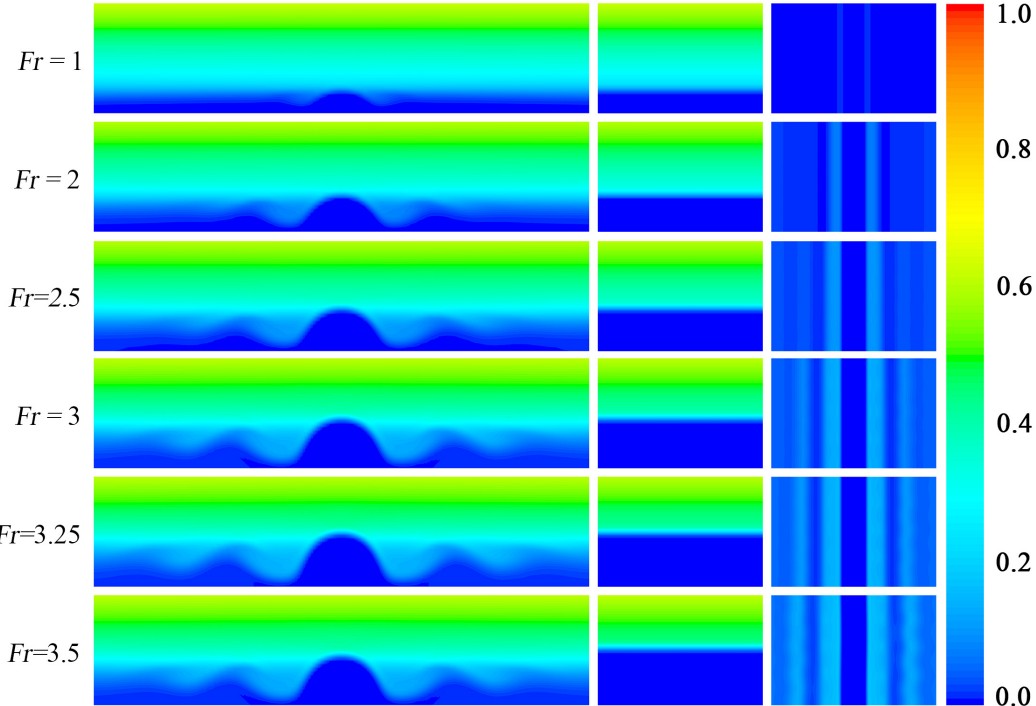

**Figure 2.** Snapshots of the temperature contours at the FDS of SPFs for several *Fr* values with *s* = 0.2, at *Y* = 0 in the *X*-*Z* plane (first column), *X* = 0 in the *Y*-*Z* plane (second column), and *z* = 0.5$z_{m,i}$ in the *X*-*Y* plane (third column), respectively.

### 3.3. Effect of s

To investigate the influence of *s*, which is the dimensionless stratification number defined by Equation (5), Figure 3 illustrates the temperature contours of five SPFs at the FDS. Similar behavior to that in Figures 1 and 2 can be identified, as fountains stay symmetric

and the mixing between the fountain and fluid is minimal. The intrusion again increases significantly compared to the fountain height, especially for a large $s$. $z_m$ reduces with increasing $s$ because of the stabilizing ability of the stratification as discussed in [10,16,17], indicating that $s$ has a notable influence on the MFPH along with the intrusion height.

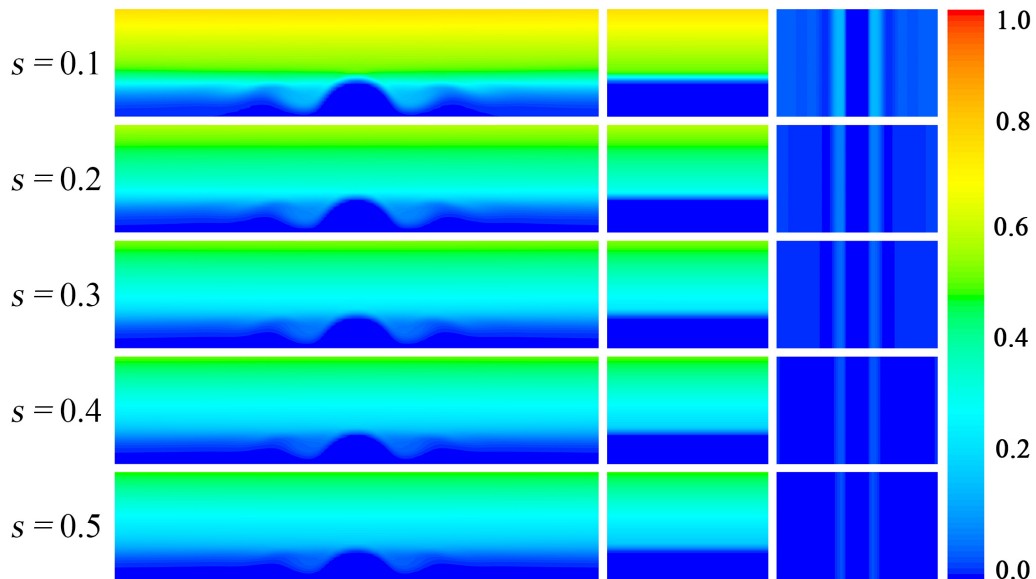

**Figure 3.** Snapshots of the temperature contours at the FDS of SPFs for several $s$ values with $Fr = 2$, at $Y = 0$ in the $X$-$Z$ plane (first column), $X = 0$ in the $Y$-$Z$ plane (second column), and $z = 0.5z_{m,i}$ in the $X$-$Y$ plane (third column).

## 4. Quantitative Analysis

### 4.1. MFPH ($z_m$)

A sample of the development of the numerically obtained $z_m$ and the corresponding velocity $v_m$, which is made dimensionless by $W_0$, are shown in Figure 4 for the SPF at $Fr = 2$ and $s = 0.1$. From Figure 4a, it can be seen that there is a continuous rise of fountain height until about $\tau = 10$, when the fountain reaches its initial MFPH, $z_{m,i}$. At this time ($\tau_{m,i}$), as expected, $v_m$ decreases to zero for the first time since the commencement of the fountain as presented in Figure 4b. Then, after a slight fall, $z_m$ rises and this growth continues at very small rates, which vary with time, and the flow is at the FDS, as can be seen in Figure 4b. This is distinctly different from the behavior of an asymmetric PF that when the fountain height reaches $z_{m,i}$, after a transition period, $z_m$ oscillates around an almost fixed value ($z_{m,a}$) at its FDS as shown in [10,16]. The observed continuous increase of the MFPH in an SPF is largely caused by the intrusion which continuously diminishes with time at the FDS, leading to reduced buoyant force experienced by the fountain fluid. It is important to quantify the MFPH at the FDS. Therefore, it was decided to find the time-averaged value of $z_m$ over a specific time period (the 'averaging period'), denoted as $z_{m,a}$, along with the MFPH at the commencement of the time averaging period, $z_{m,s}$, as depicted in Figure 4a. The time-averaged velocity $v_{m,a}$ during the averaging period is apparently the appropriate parameter to quantify the extent of the variation of $z_m$ at the FDS. In this study, for consistency, the time instant for $z_{m,s}$ is at $\tau = 100$, while the averaging period for both $z_{m,a}$ and $v_{m,a}$ is over $100 \leq \tau \leq 900$ for all fountains considered.

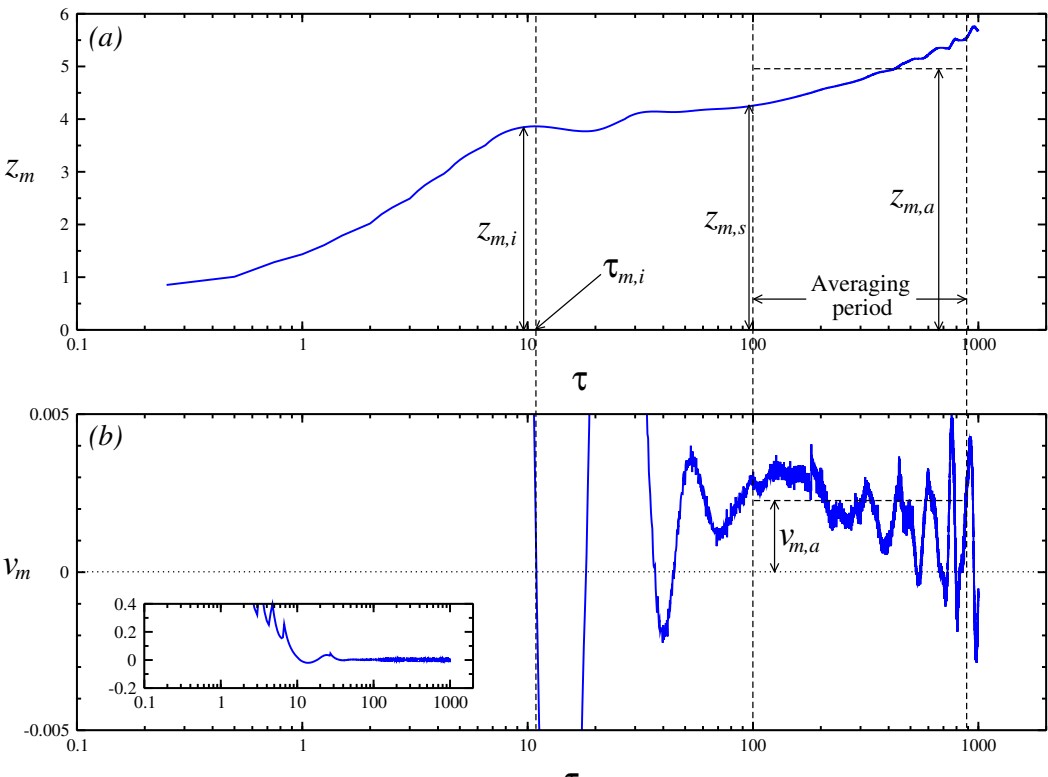

**Figure 4.** Time series of $z_m$ (**a**) and $v_m$ (**b**) for the SPF of $Fr = 2$ and $s = 0.1$, where $z_{m,i}$ is the initial MFPH and $\tau_{m,i}$ is the time instant when $z_m = z_{m,i}$, $z_{m,s}$ is the MFPH at the commencement time of the period for the time averaging of $z_m$ at the FDS which gives the time-averaged MFPH $z_{m,a}$. All heights, times and velocity are made dimensionless by $X_0$, $X_0/W_0$ and $W_0$, respectively. The averaging period for both $z_{m,a}$ and $v_{m,a}$ is $100 \le \tau \le 900$.

The numerical obtained $z_m$ and $v_m$ for five SPFs with five $Fr$ values at $s = 0.1$ and five SPFs with five $s$ values at $Fr = 2$ are shown in Figure 5. It is clearly seen that both $Fr$ and $s$ have significant effects on $z_m$ and $v_m$. As expected, $z_m$ increases when $Fr$ becomes larger, but it decreases when $s$ increases, as the corresponding negative buoyancy increases too, which leads consequently to smaller $\tau_{m,i}$, $z_{m,s}$, $z_{m,a}$ and $v_{m,a}$, as will be shown subsequently.

The effects of $Fr$ and $s$ on $z_{m,i}$ are illustrated by the numerical results shown in Figure 6 for all SPFs considered in the present study. As shown in Figure 6a, for each $s$ value, $z_{m,i}$ increases when $Fr$ becomes larger, which can be quantified by an approximately quadratic correlation, owing to the increased momentum flux of the fountain. It is also observed that for each $Fr$, the increase of $s$ results in reduced $z_{m,i}$, as also shown in Figure 6b, resulting from a stronger negative buoyancy.

For a weak SPF in an LSF, Lin and Armfield [2] asserted that momentum and buoyancy fluxes, kinematic viscosity of fluid, and ambient stratification together form a complete parametrization of the MFPH, and they derived the scaling relation for $z_m$ through a dimensional analysis, as shown below,

$$z_{m,i} \sim Fr^{\frac{2}{3}(2+2b-c)} s^b Re^{-c}, \tag{6}$$

where the constants $b$ and $c$ can be obtained empirically with experimental or numerical results. They confirmed this scaling relation for weak SPFs with a series of numerical simulations with varied $Fr$, $Re$ and $s$ values. For the weak SPFs considered in the present study, with $Re$ fixed at $Re = 100$, it is impossible to obtain the value of $c$, but apparently, the overall effects of $Fr$ and $s$ on $z_{m,i}$ can be represented by $Fr^a s^b$, where $a$ is another constant which can be determined empirically.

With the numerical results obtained for all fountains considered here, a multi-variable regression analysis gives $a = 0.79$ and $b = -0.178$ for $z_{m,i}$, and the following correlation is obtained to quantify the overall effects of $Fr$ and $s$ on $z_{m,i}$,

$$\begin{aligned} z_{m,i} &= 0.22(Fr^{0.79}s^{-0.178})^2 + 0.46Fr^{0.79}s^{-0.178} + 1.106 \\ &= 0.22Fr^{1.58}s^{-0.356} + 0.46Fr^{0.79}s^{-0.178} + 1.106, \end{aligned} \tag{7}$$

with the regression constant of $R^2 = 0.9947$. This indicates that $Fr^{0.79}s^{-0.178}$ quantifies the overall effects of $Fr$ and $s$ on $z_{m,i}$ very well, as also shown in Figure 6c.

Nevertheless, it was noted from the numerical simulations that among all fountain cases considered, there are four cases at high $Fr$ and $s$ values that become asymmetric after $z_m$ attains $z_{m,i}$, although only slightly. These cases are those at $Fr = 4.5$ and $Fr = 5$ with $s = 0.4$ and at $Fr = 4.85$ and $Fr = 5$ with $s = 0.5$. Hence, the results of these four cases should be excluded in the multi-variable regression analysis to obtain the correlation to quantify the combined effects of $Fr$ and $s$. With the exclusion of the results of these four cases, it is found that $a$ and $b$ have very small changes from those without the exclusions, i.e., $a$ changes from 0.79 to 0.768, and $b$ changes from $-0.178$ to $-0.188$. The following corresponding correlation is then obtained with $Fr^{0.768}s^{-0.188}$,

$$\begin{aligned} z_{m,i} &= 0.214(Fr^{0.768}s^{-0.188})^2 + 0.535Fr^{0.768}s^{-0.188} + 0.99 \\ &= 0.214Fr^{1.536}s^{-0.376} + 0.535Fr^{0.768}s^{-0.188} + 0.99, \end{aligned} \tag{8}$$

with $R^2 = 0.9963$, indicating that $Fr^{0.768}s^{-0.188}$ provides a slightly better representation of the overall effects of both $Fr$ and $s$ on $z_{m,i}$, as shown in Figure 6d.

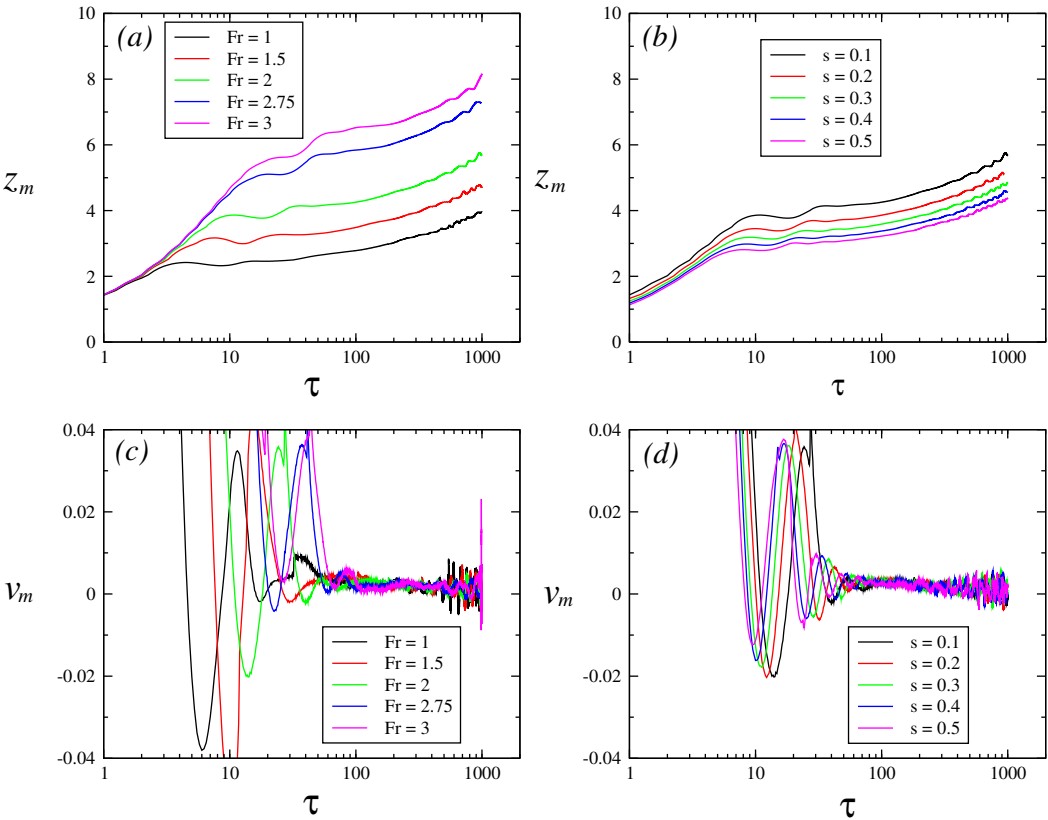

**Figure 5.** Time series of $z_m$ (**a**) and $v_m$ (**c**) for the five SPFs with five $Fr$ values at $s = 0.1$, and that of $z_m$ (**b**) and $v_m$ (**d**) for the five SPFs with five $s$ values at $Fr = 2$, respectively.

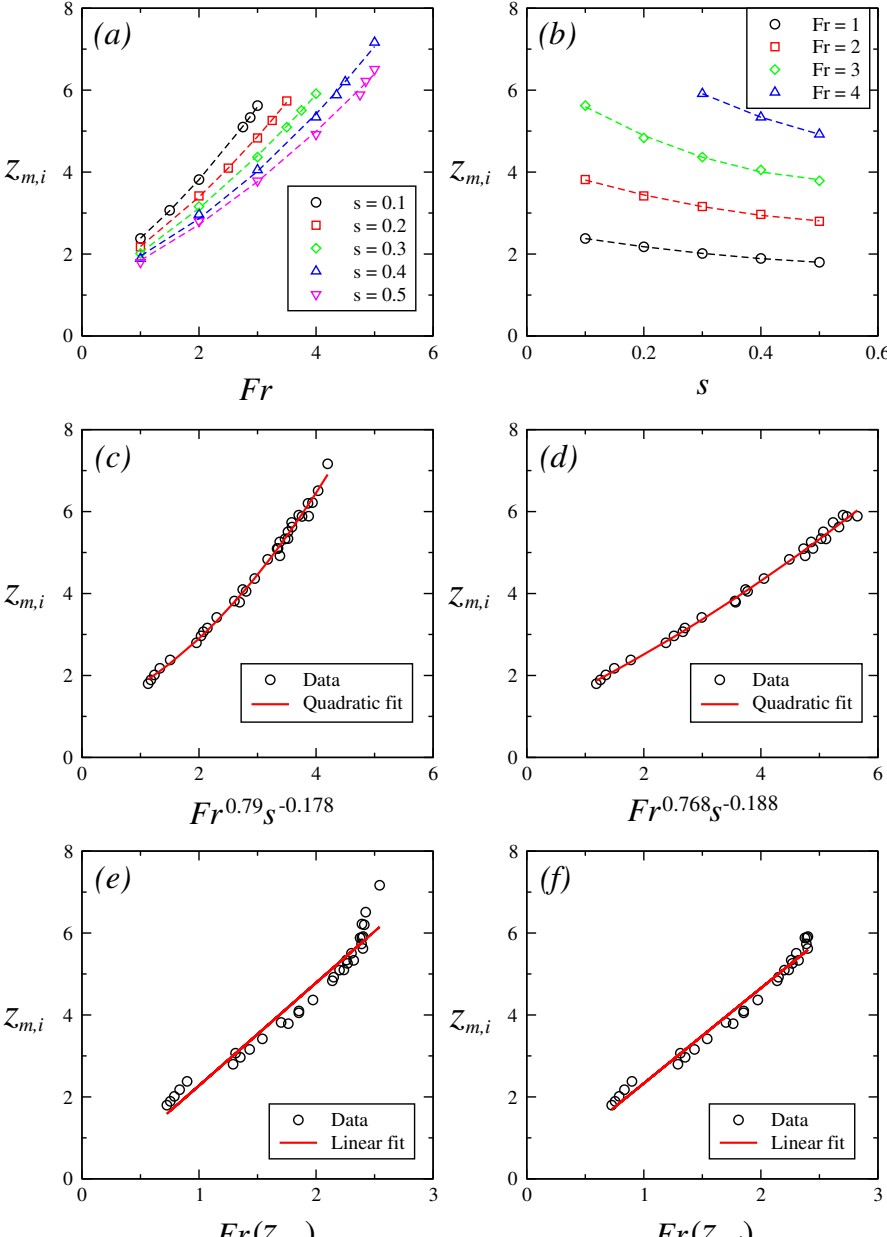

**Figure 6.** (**a**) $z_{m,i}$ plotted against $Fr$ at $s = 0.1, 0.2, 0.3, 0.4$ and $0.5$, (**b**) $z_{m,i}$ plotted against $s$ at $Fr = 1,$ 2, 3 and 4, (**c**) $z_{m,i}$ plotted against $Fr^{0.79}s^{-0.178}$ over $0.1 \le s \le 0.5$ and $1 \le Fr \le 5$, without exclusions, (**d**) $z_{m,i}$ plotted against $Fr^{0.768}s^{-0.188}$ over $0.1 \le s \le 0.5$ and $1 \le Fr \le 5$, with exclusions, (**e**) $z_{m,i}$ plotted against $Fr(z_{m,i})$ over $0.1 \le s \le 0.5$ and $1 \le Fr \le 5$, without exclusions, (**f**) $z_{m,i}$ plotted against $Fr(z_{m,i})$ over $0.1 \le s \le 0.5$ and $1 \le Fr \le 5$, with exclusions, respectively.

A local Froude number at height $z$ can be introduced, which is defined as

$$Fr(z) = \frac{W_0}{\sqrt{g\beta X_0(T_0 - T_{a,Z})}}. \tag{9}$$

It can be found that

$$Fr(z) = \frac{Fr}{\sqrt{1 + sz}}. \tag{10}$$

It is appropriate to assume that the effect of $s$ is incorporated in $Fr(z)$. Hence, it is expected that $z_{m,i}$ depends on $Fr(z_{m,i})$ only and the scaling relation $z_{m,i} \sim Fr$ developed for weak SPFs in homogeneous ambient fluids by Lin and Armfield [6] and others should

be applicable here as well if $Fr$ is replaced by $Fr(z_{m,i})$, as validated by the numerical results shown in Figure 6e,f. From the results for all fountains considered, the following linear correlation is obtained with a linear regression analysis,

$$z_{m,i} = 2.509 Fr(z_{m,i}) - 0.233, \tag{11}$$

with $R^2 = 0.9521$, indicating that the scaling relation $z_{m,i} \sim Fr(z_{m,i})$ is applicable well for the majority of the fountains considered, as shown in Figure 6e. Similarly, the results for the four cases mentioned above should be excluded, which changes the correlation to be

$$z_{m,i} = 2.324 Fr(z_{m,i}) + 0.007, \tag{12}$$

with $R^2 = 0.9793$. It is clearly seen that the scaling relation $z_{m,i} \sim Fr(z_{m,i})$ is applicable very well for all fountains with the exclusion of the four fountains, as indicated in Figure 6f.

As $z_{m,s}$ and $z_{m,a}$ are essentially the parameter as $z_{m,i}$ to represent the MFPH and differ from $z_{m,i}$ only in that they quantify the MFPH at different times, it is expected that the characteristics of $z_{m,s}$ and $z_{m,a}$ should be the same as that of $z_{m,i}$ and all conclusions obtained above for $z_{m,i}$ are applicable to $z_{m,i}$ too. However, there are quantitative differences in the correlations for $z_{m,s}$ and $z_{m,a}$, as shown in Figures 7 and 8. With the numerical results obtained for all fountains considered, the correlations are obtained for $z_{m,s}$ and $z_{m,a}$, both without the exclusion of any fountains and with the exclusion of the four fountains mentioned above, and these are listed in Table 1. As shown in Figures 7 and 8, all these correlations indicate that, similar to that for $z_{m,i}$, the respective $Fr^a s^b$ obtained quantifies the overall effects of $Fr$ and $s$ on $z_{m,s}$ and $z_{m,a}$ very well, with those with the exclusion of the four fountains providing a slightly better representation of the overall effects of both $Fr$ and $s$.

**Table 1.** Numerical obtained empirical correlations for $z_{m,s}$ and $z_{m,a}$ with and without the exclusions.

| Correlation | Exclusion? | $R^2$ |
|---|---|---|
| $z_{m,s} = 0.316 Fr^{1.49} s^{-0.342} + 0.392 Fr^{0.745} s^{-0.171} + 1.434$ | No | 0.9938 |
| $z_{m,s} = 0.411 Fr^{1.46} s^{-0.356} + 0.022 Fr^{0.73} s^{-0.178} + 1.754$ | Yes | 0.9956 |
| $z_{m,s} = 2.855 Fr(z_{m,s}) - 0.0873$ | No | 0.9578 |
| $z_{m,s} = 2.696 Fr(z_{m,s}) + 0.113$ | Yes | 0.9684 |
| $z_{m,a} = 0.428 Fr^{1.318} s^{-0.328} + 0.514 Fr^{0.659} s^{-0.164} + 1.682$ | No | 0.9958 |
| $z_{m,a} = 0.523 Fr^{1.296} s^{-0.338} + 0.179 Fr^{0.648} s^{-0.169} + 1.949$ | Yes | 0.9967 |
| $z_{m,a} = 2.940 Fr(z_{m,a}) + 0.585$ | No | 0.9762 |
| $z_{m,a} = 2.814 Fr(z_{m,a}) + 0.738$ | Yes | 0.9830 |

A similar analysis can be made on $\tau_{m,i}$, which is another key parameter. The effects of $Fr$ and $s$ on $\tau_{m,i}$ are indicated by the numerical results shown in Figure 9 for all fountains considered. Similar to that for $z_{m,i}$, as shown in Figure 9a, for each $s$ value, $\tau_{m,i}$ increases with increasing $Fr$, which can also be quantified by an approximately quadratic correlation because of the larger momentum flux, and for each $Fr$ value, the increase of $s$ results in reduced $\tau_{m,i}$, which can be seen from Figure 9b as well because of the increased negative buoyancy.

Similar to that for $z_{m,i}$, the overall effects of $Fr$ and $s$ on $\tau_{m,i}$ can be quantified by $Fr^a s^b$. However, apparently, it is expected that the values of $a$ and $b$ obtained from the numerical results should be significantly different from those for $z_{m,i}$. These are confirmed by the results presented in Figure 9c–f. A multi-variable regression analysis of the results for $\tau_{m,i}$ from all fountains considered shows that $Fr^{1.273} s^{-0.229}$ quantifies the overall effects of $Fr$ and $s$ on $\tau_{m,i}$ well, with the following correlation obtained when all fountains are included,

$$\begin{aligned} \tau_{m,i} &= 0.209 (Fr^{1.273} s^{-0.229})^2 + 1.879 Fr^{1.273} s^{-0.229} + 2.588 \\ &= 0.209 Fr^{2.546} s^{-0.458} + 1.879 Fr^{1.273} s^{-0.229} + 2.588, \end{aligned} \tag{13}$$

with $R^2 = 0.9752$, as shown in Figure 9c. Similarly, the results of the four fountains mentioned above should be excluded in the regression analysis, and it is found that the obtained $Fr^{1.225}s^{-0.252}$ provides a slightly better representation of the overall effects of $Fr$ and $s$ on $\tau_{m,i}$, with the following correlation obtained when the four fountains are excluded,

$$
\begin{aligned}
\tau_{m,i} &= 0.106(Fr^{1.225}s^{-0.252})^2 + 2.897Fr^{1.225}s^{-0.252} + 0.855 \\
&= 0.106Fr^{2.45}s^{-0.504} + 2.897Fr^{1.225}s^{-0.252} + 0.855,
\end{aligned}
\tag{14}
$$

with $R^2 = 0.9827$, as shown in Figure 9d.

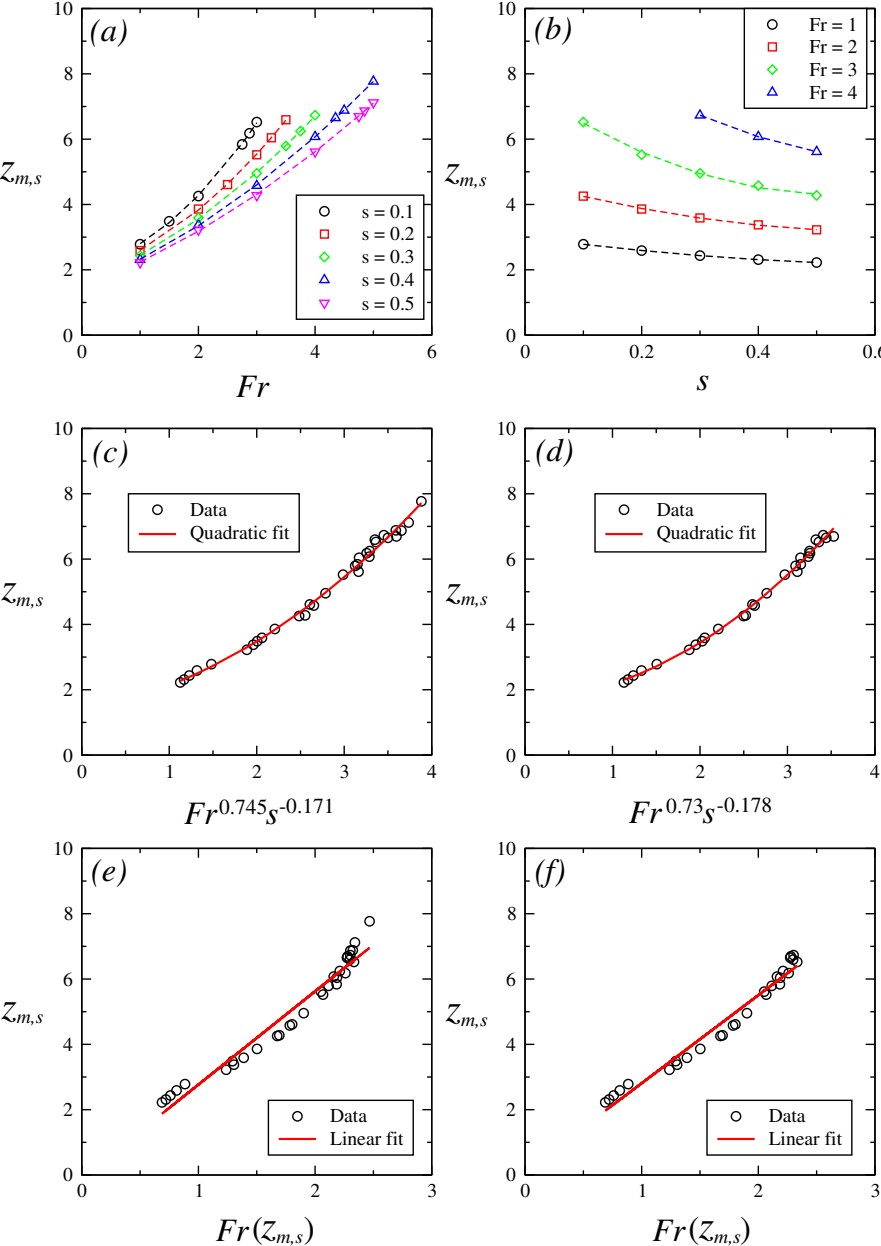

**Figure 7.** (a) $z_{m,s}$ plotted against $Fr$ at $s = 0.1, 0.2, 0.3, 0.4$ and 0.5, (b) $z_{m,s}$ plotted against $s$ at $Fr = 1,$ 2, 3 and 4, (c) $z_{m,s}$ plotted against $Fr^{0.745}s^{-0.171}$ over $0.1 \leq s \leq 0.5$ and $1 \leq Fr \leq 5$, without exclusions, (d) $z_{m,s}$ plotted against $Fr^{0.73}s^{-0.178}$ over $0.1 \leq s \leq 0.5$ and $1 \leq Fr \leq 5$, with exclusions, (e) $z_{m,s}$ plotted against $Fr(z_{m,s})$ over $0.1 \leq s \leq 0.5$ and $1 \leq Fr \leq 5$, without exclusions, (f) $z_{m,s}$ plotted against $Fr(z_{m,s})$ over $0.1 \leq s \leq 0.5$ and $1 \leq Fr \leq 5$, with exclusions, respectively.

For weak symmetric fountains in homogeneous ambient fluids, Lin and Armfield [6,19] show that $\tau_{m,i} \sim Fr^2$. With the effect of $s$ incorporated in $Fr(z)$, similar to that for $z_{m,i}$, it is expected that $\tau_{m,i}$ depends on $Fr(z_{m,i})$ only and the scaling relation $\tau_{m,i} \sim Fr^2$ developed by Lin and Armfield [6,19] should be applicable here as well if $Fr$ is replaced by $Fr(z_{m,i})$. This is validated by the results shown in Figure 9e,f. From the results for all fountains considered, the following correlation is obtained with a regression analysis,

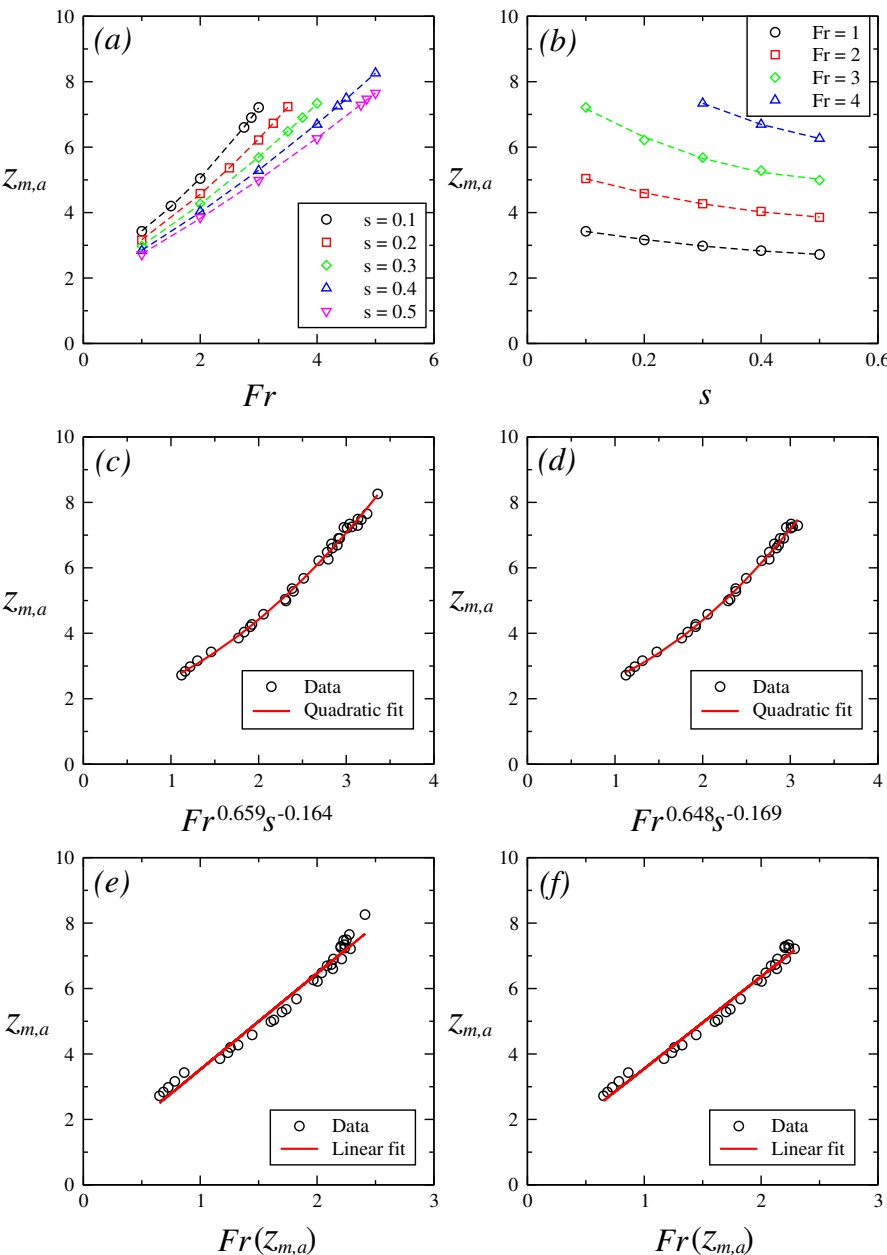

**Figure 8.** (**a**) $z_{m,a}$ plotted against $Fr$ at $s = 0.1$, $0.2$, $0.3$, $0.4$ and $0.5$, (**b**) $z_{m,a}$ plotted against $s$ at $Fr = 1$, 2, 3 and 4, (**c**) $z_{m,a}$ plotted against $Fr^{0.659}s^{-0.164}$ over $0.1 \leq s \leq 0.5$ and $1 \leq Fr \leq 5$, without exclusions, (**d**) $z_{m,a}$ plotted against $Fr^{0.648}s^{-0.169}$ over $0.1 \leq s \leq 0.5$ and $1 \leq Fr \leq 5$, with exclusions, (**e**) $z_{m,a}$ plotted against $Fr(z_{m,a})$ over $0.1 \leq s \leq 0.5$ and $1 \leq Fr \leq 5$, without exclusions, (**f**) $z_{m,a}$ plotted against $Fr(z_{m,a})$ over $0.1 \leq s \leq 0.5$ and $1 \leq Fr \leq 5$, with exclusions, respectively.

$$\tau_{m,i} = 10.246 Fr(z_{m,i})^2 - 17.263 Fr(z_{m,i}) + 13.198, \tag{15}$$

with $R^2 = 0.9353$, indicating that the scaling relation $\tau_{m,i} \sim Fr(z_{m,i})^2$ is applicable well for the majority of the fountains considered, as shown in Figure 9e. Similarly, the results

for the four cases mentioned above should be excluded in the regression analysis, which produces the following correlation,

$$\tau_{m,i} = 5.944 Fr(z_{m,i})^2 - 4.958 Fr(z_{m,i}) + 5.611, \tag{16}$$

with $R^2 = 0.9879$. This shows that the scaling relation $\tau_{m,i} \sim Fr(z_{m,i})^2$ is applicable very well for all fountains with the exclusion of the four fountains, as shown in Figure 9f.

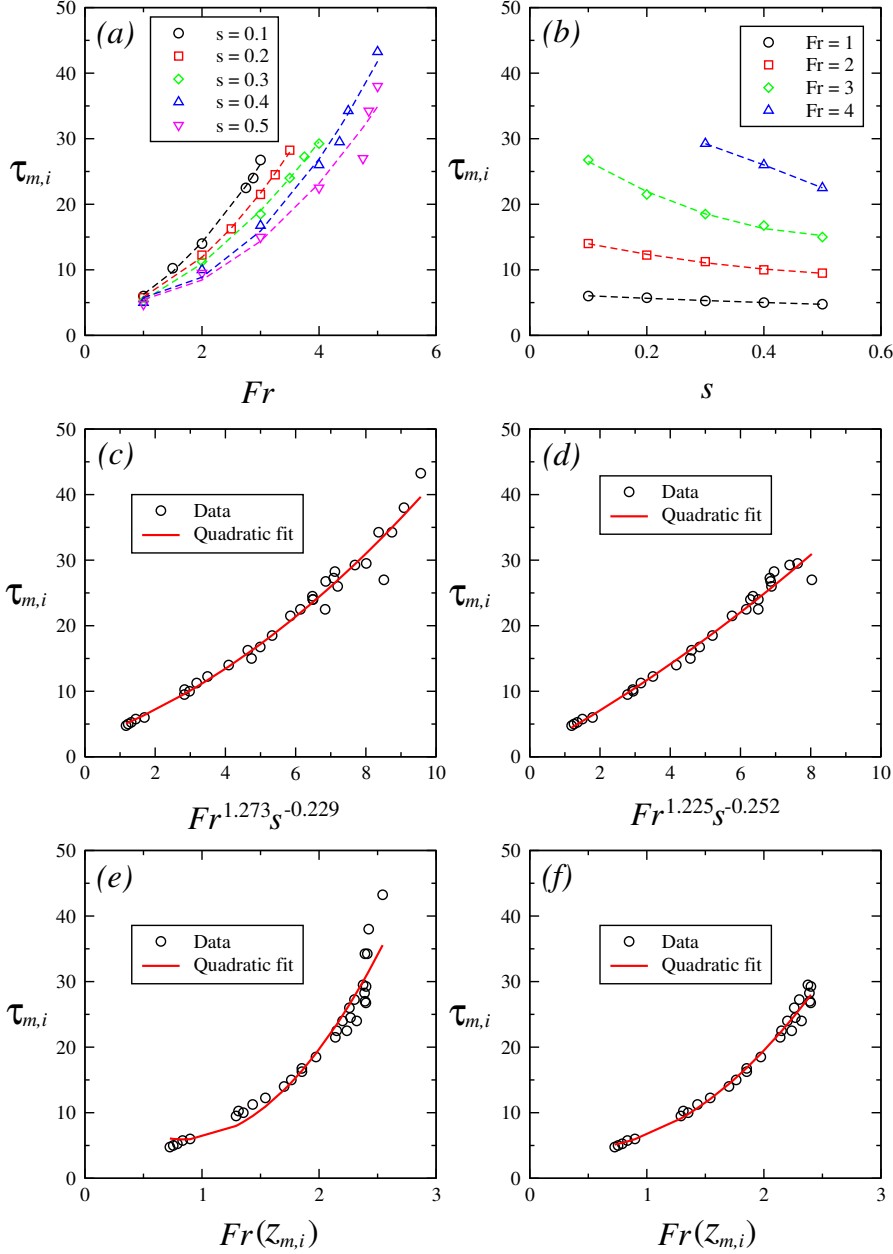

**Figure 9.** (**a**) $\tau_{m,i}$ plotted against $Fr$ at $s = 0.1, 0.2, 0.3, 0.4$ and $0.5$, (**b**) $\tau_{m,i}$ plotted against $s$ at $Fr = 1$, $2, 3$ and $4$, (**c**) $\tau_{m,i}$ plotted against $Fr^{1.273}s^{-0.229}$ over $0.1 \leq s \leq 0.5$ and $1 \leq Fr \leq 5$, without exclusions, (**d**) $\tau_{m,i}$ plotted against $Fr^{1.225}s^{-0.252}$ over $0.1 \leq s \leq 0.5$ and $1 \leq Fr \leq 5$, with exclusions, (**e**) $\tau_{m,i}$ plotted against $Fr(z_{m,i})$ over $0.1 \leq s \leq 0.5$ and $1 \leq Fr \leq 5$, without exclusions, (**f**) $\tau_{m,i}$ plotted against $Fr(z_{m,i})$ over $0.1 \leq s \leq 0.5$ and $1 \leq Fr \leq 5$, with exclusions, respectively.

Figure 10 presents the numerical results for $v_{m,a}$, which quantifies the increase rate of $z_m$ with time during the averaging period at the FDS. Figure 10a shows that for each $s$, $Fr$ only significantly affects $v_{m,a}$ when $Fr$ is no more than 3, with $v_{m,a}$ increasing significantly

when $Fr$ becomes larger. However, beyond $Fr \approx 3$, the effect of $Fr$ on $v_{m,a}$ is very small. It is also observed that the variation of $s$ results in noticeable changes of $v_{m,a}$ when $Fr$ is no more than 3, with a larger $s$ value producing a reduced value of $v_{m,a}$, but beyond $Fr \approx 3$, the effect of $s$ on $v_{m,a}$ becomes negligible, as shown in Figure 10b.

A multi-variable regression analysis of the results for $v_{m,s}$ from all fountains considered shows that $Fr^{0.22}s^{-0.155}$ provides a reasonable representation of the overall effects of $Fr$ and $s$ on $v_{m,a}$, and the following correlation is obtained when all fountains are included,

$$
\begin{aligned}
v_{m,a} &= -0.0011(Fr^{0.22}s^{-0.155})^2 + 0.0043Fr^{0.22}s^{-0.155} - 0.0023 \\
&= -0.0011Fr^{0.44}s^{-0.31} + 0.0043Fr^{0.22}s^{-0.155} - 0.0023,
\end{aligned} \tag{17}
$$

with $R^2 = 0.8036$, as shown in Figure 10c. Similarly, the results of the four fountains mentioned above should be excluded in the regression analysis, and it is found that the obtained $Fr^{0.238}s^{-0.147}$ provides a slightly better representation of the overall effects of $Fr$ and $s$ on $v_{m,a}$, with the following correlation obtained when the four fountains are excluded,

$$
\begin{aligned}
v_{m,a} &= -0.0012(Fr^{0.238}s^{-0.147})^2 + 0.0046Fr^{0.238}s^{-0.147} - 0.0025 \\
&= -0.0012Fr^{0.476}s^{-0.294} + 0.0046Fr^{0.238}s^{-0.147} - 0.0025,
\end{aligned} \tag{18}
$$

with $R^2 = 0.8485$, as shown in Figure 10d.

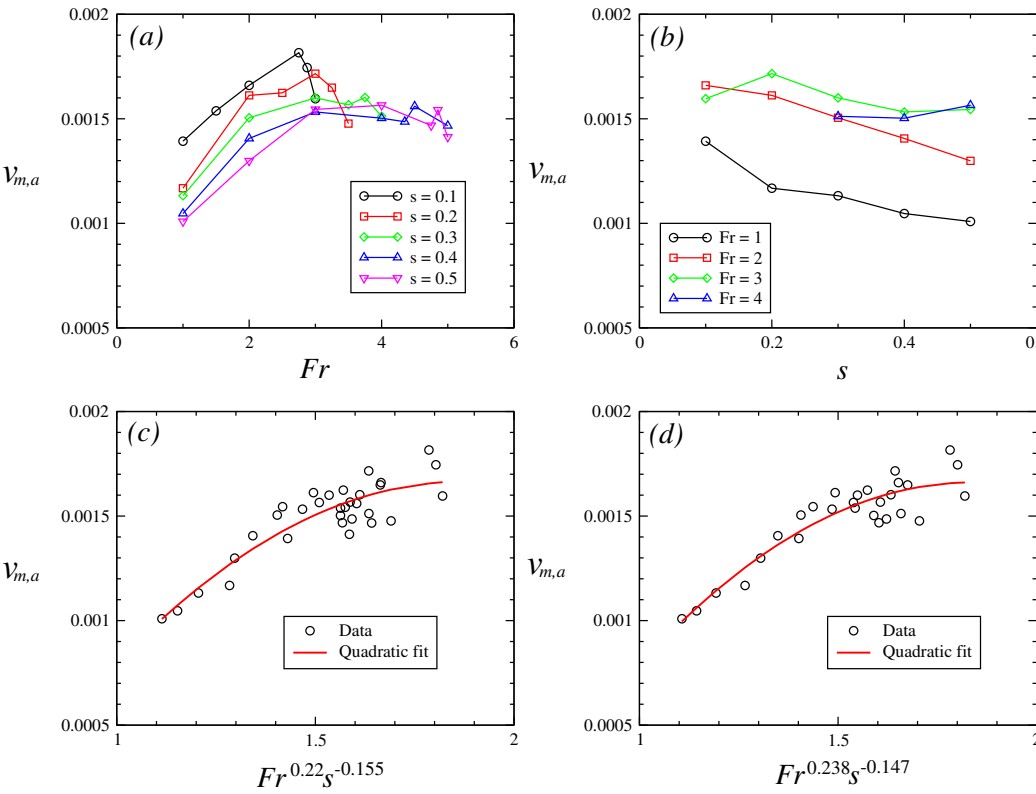

**Figure 10.** (**a**) $v_{m,a}$ plotted against $Fr$ at $s = 0.1$, 0.2, 0.3, 0.4 and 0.5, (**b**) $v_{m,a}$ plotted against $s$ at $Fr = 1$, 2, 3 and 4, (**c**) $v_{m,a}$ plotted against $Fr^{0.22}s^{-0.155}$ over $0.1 \le s \le 0.5$ and $1 \le Fr \le 5$, without exclusions, (**d**) $v_{m,a}$ plotted against $Fr^{0.238}s^{-0.147}$ over $0.1 \le s \le 0.5$ and $1 \le Fr \le 5$, with exclusions, respectively.

### 4.2. Intrusion

As explained earlier, intrusion is another key feature of the fountain behavior of an SPF in LSF. Due to the dense fluid injected, the intrusion formation can be observed to be on the bottom surface only after the downflow reaches the bottom surface and subsequently moves outward along the bottom floor. Therefore, intrusion development and its movement alter ambient fluid stratification, leading to a reduced negative buoyant force applied to

the fountain. This effect is especially notable for large $s$ and small $Fr$ values that observed MFPH is considerably smaller.

Figure 11 presents an example of the intrusion for the SPF at $Fr = 2$ and $s = 0.1$ with its temperature contours and outer boundary region. The major parameters characterizing the intrusion behavior include $x_{int}$ and $u_{int}$, which are the instantaneous dimensionless intrusion front distance away from $X = 0$ and the corresponding dimensionless velocity, as depicted in Figure 11b.

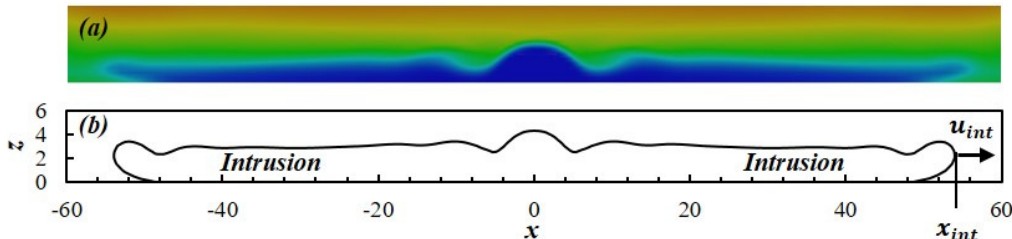

**Figure 11.** (**a**) The temperature contour and (**b**) the outer boundary of the intrusion region at $Y = 0$ in the $X$-$Z$ plane, which is the iso-temperature curve at $T(Z) = T_0 - 1\%(T_{a,0} - T_0)$, for the SPF at $Fr = 2$ and $s = 0.1$, where $x_{int}$ and $u_{int}$ are the instantaneous intrusion front distance away from $X = 0$ and the corresponding velocity, which are made dimensionless by $X_0$ and $W_0$, respectively.

The numerically obtained time series of $x_{int}$ and $u_{int}$ of the SPF at $Fr = 2$ and $s = 0.2$ are presented in Figure 12. It is seen that the evolution of $x_{int}$ experiences three distinct stages: the initial stage (Stage 1) from the formation of the intrusion until $u_{int}$ attains the maximum, in which $u_{int}$ increases continually; subsequently, $u_{int}$ reduces monotonically for a period of time (Stage 2); and eventually, the intrusion is at the FDS (Stage 3) in which $u_{int}$ continually decreases but at the rates that are much smaller than those in Stage 2. As the instantaneous values of $x_{int}$ at different times are determined automatically in the code, the locations of $x_{int}$, which is the furthermost point of the intrusion front, may be at different heights. These results in the fluctuations in the time series of $u_{int}$ due to the very small time step used and the very long whole time duration considered, as shown in Figure 12b. To smooth these fluctuations, the moving average of $u_{int}$, denoted as $\bar{u}_{int}$, is considered to be a better representation of $u_{int}$. The time series of $\bar{u}_{int}$ is presented in Figure 12c. In Figure 12d, the time series of the corresponding rate of $\bar{u}_{int}$ changing with time, i.e., the acceleration $\bar{a}_{int}$, is shown. For all fountains considered, the moving average interval of 5 in dimensionless time is used.

It is noted that these three stages (Stages 1, 2 and 3) for the evolution of $x_{int}$ are very similar to the three regimes for the development of a pure gravity current, i.e., the wall jet regime, the buoyancy–inertial regime, and the buoyancy–viscosity regime, which are distinguished by the respective dominating forces [20]. In the wall jet regime, the flow behaves as a plane wall jet with the dominant momentum, which is followed by the second regime where buoyancy becomes the driving force which is balanced by the inertial force; and gradually, the inertial force decreases and the total viscous force caused by the interfacial shear stress between the gravity current and the ambient fluid and the bottom shear stress increases, eventually evolving into the buoyancy–viscosity regime, where the buoyancy force is balanced by the viscous drag force [20].

The effects of $Fr$ and $s$ on $x_{int}$ and $\bar{u}_{int}$ are demonstrated by the numerical results presented in Figure 13 for five SPFs with varying $Fr$ at $s = 0.1$ and five SPFs with varying $s$ at $Fr = 2$. The results show that the times series of $x_{int}$ and $\bar{u}_{int}$ are very similar for all fountains, although they differ quantitatively. $Fr$ is found to significantly affect both $x_{int}$ and $\bar{u}_{int}$, whereas the effect of $s$ is much smaller, particularly at the FDS (Stage 3).

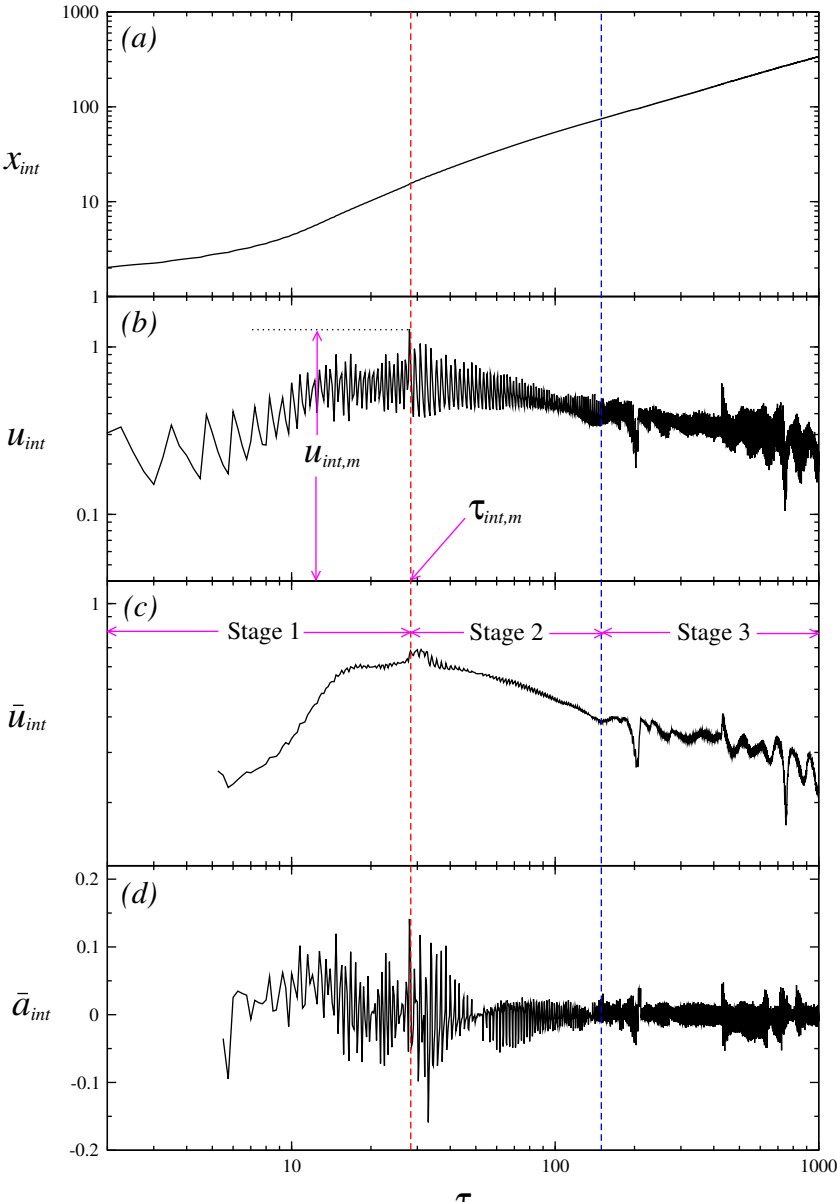

**Figure 12.** Time series of (**a**) $x_{int}$, (**b**) $u_{int}$, (**c**) $\bar{u}_{int}$, and (**d**) $\bar{a}_{int}$ for the SPF at $Fr = 2$ and $s = 0.1$, where $u_{int,m}$ is the instantaneous maximum horizontal velocity of the intrusion front and $\tau_{int,m}$ is the time instant when $u_{int} = u_{int,m}$, $\bar{u}_{int}$ is the moving average of $u_{int}$ with the averaging period of 5 (dimensionless time), and $\bar{a}_{int}$ is the rate of $\bar{u}_{int}$ changing with time (i.e., the acceleration of $x_{int}$). $x_{int}$, $u_{int}$, $\bar{u}_{int}$, and $\bar{a}_{int}$ are made dimensionless by $X_0$, $V_0$, $V_0$, and $X_0/V_0^2$ and $V_0$, respectively.

The effects of $Fr$ and $s$ on $\bar{u}_{int,m}$, which is the moving average of the maximum horizontal velocity of the intrusion front, are shown in Figure 14 for all SPFs considered. As shown in Figure 14a, for each $s$, $\bar{u}_{int,m}$ reduces significantly when $Fr$ becomes larger, which can be quantified by a power-law correlation. However, the effect of $s$ on $\bar{u}_{int,m}$ is negligible, as all data with different $s$ values are essentially on the same power-law curve. This is also clearly demonstrated in Figure 14b which shows that for each $Fr$ value, $\bar{u}_{int,m}$ almost does not vary, particularly when $Fr$ is beyond 1.

Similar to that for $z_m$, the overall effects of $Fr$ and $s$ on $\bar{u}_{int,m}$ can also be quantified by $Fr^a s^b$. A multi-variable regression analysis of the numerical results for $\bar{u}_{int,m}$ from all

fountains considered shows that $Fr^{-0.603}s^{-0.013}$ quantifies the overall effects of $Fr$ and $s$ on $\bar{u}_{int,m}$ very well, with the following correlation obtained when all fountains are included,

$$\bar{u}_{int,m} = 0.9932Fr^{-0.603}s^{-0.013} + 0.0153, \qquad (19)$$

with $R^2 = 0.9925$. From this correlation, it is seen that $c = -0.013$, showing that the effect of $s$ is negligible, which is in agreement with the results presented in Figure 14a,b. Hence, it is expected that the omission of the effect of $s$ on $\bar{u}_{int,m}$ should not lead to a noticeable change of $d$. This is verified by the result presented in Figure 14c, which shows that the following power-law correlation can be obtained,

$$\bar{u}_{int,m} = 1.0404Fr^{-0.607}, \qquad (20)$$

with $R^2 = 0.9893$, where $a = -0.607$, which is almost the same as $a = -0.603$ as obtained in the correlation (19) when the effect of $s$ is included.

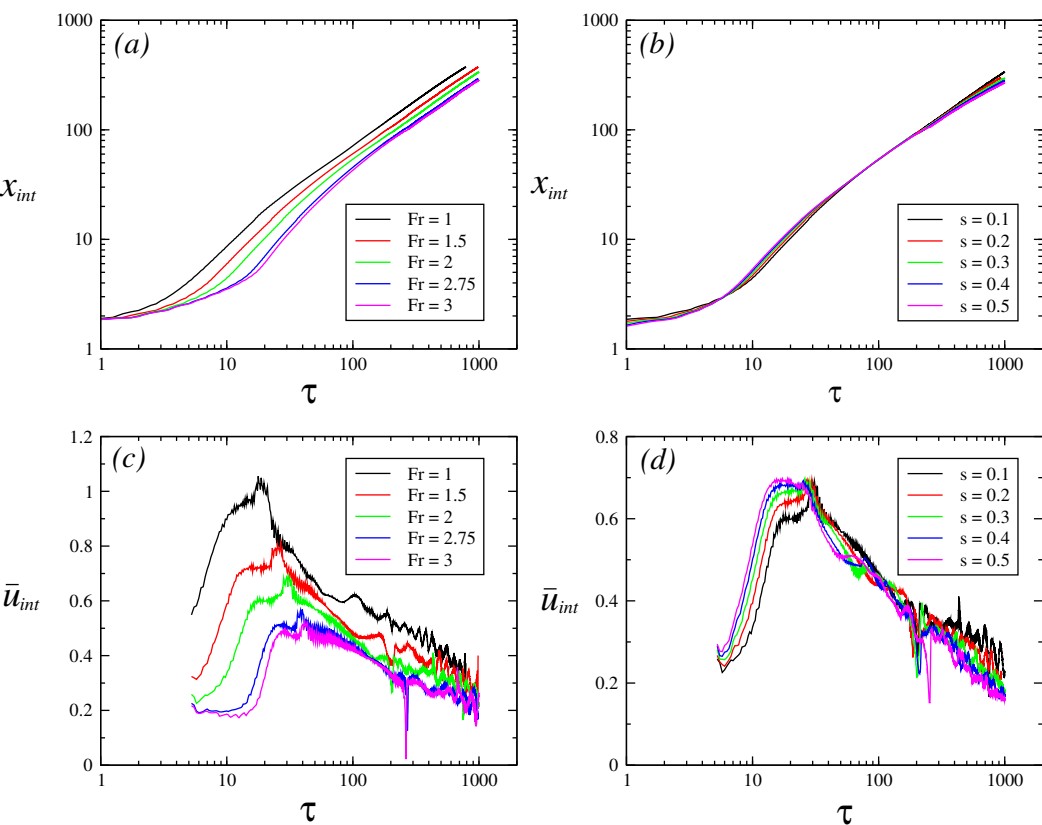

**Figure 13.** Time series of $x_{int}$ (**a**) and $\bar{u}_{int}$ (**c**) for the five SPFs with five $Fr$ values at $s = 0.1$, and the time series of $x_{int}$ (**b**) and $\bar{u}_{int}$ (**d**) for the five SPFs for the five $s$ values at $Fr = 2$, respectively.

The effects of $Fr$ and $s$ on $\tau_{int,m}$, which is the time instant when $\bar{u}_{int,m}$ attains the maximum, are demonstrated by the numerical results presented in Figure 15. As shown in Figure 15a, for each $s$ value, $\tau_{int,m}$ increases significantly when $Fr$ increases, which can be quantified by a power-law correlation. $s$ also has an effect on $\tau_{int,m}$, although not as strong as $Fr$ has, particularly when $Fr$ is larger. This is also clearly shown in Figure 15b which shows that for each $Fr$ value, $\tau_{int,m}$ decreases when $s$ increases.

Likewise, the overall effects of $Fr$ and $s$ on $\tau_{int,m}$ can also be quantified by $Fr^a s^b$. A multi-variable regression analysis of the results for $\tau_{int,m}$ from all fountains shows that $Fr^{0.695}s^{-0.14}$ quantifies the overall effects of $Fr$ and $s$ on $\tau_{int,m}$ very well, and the following correlation is obtained when all fountains are included,

$$\tau_{int,m} = 14.039Fr^{0.695}s^{-0.14} - 0.2744, \qquad (21)$$

with $R^2 = 0.9935$.

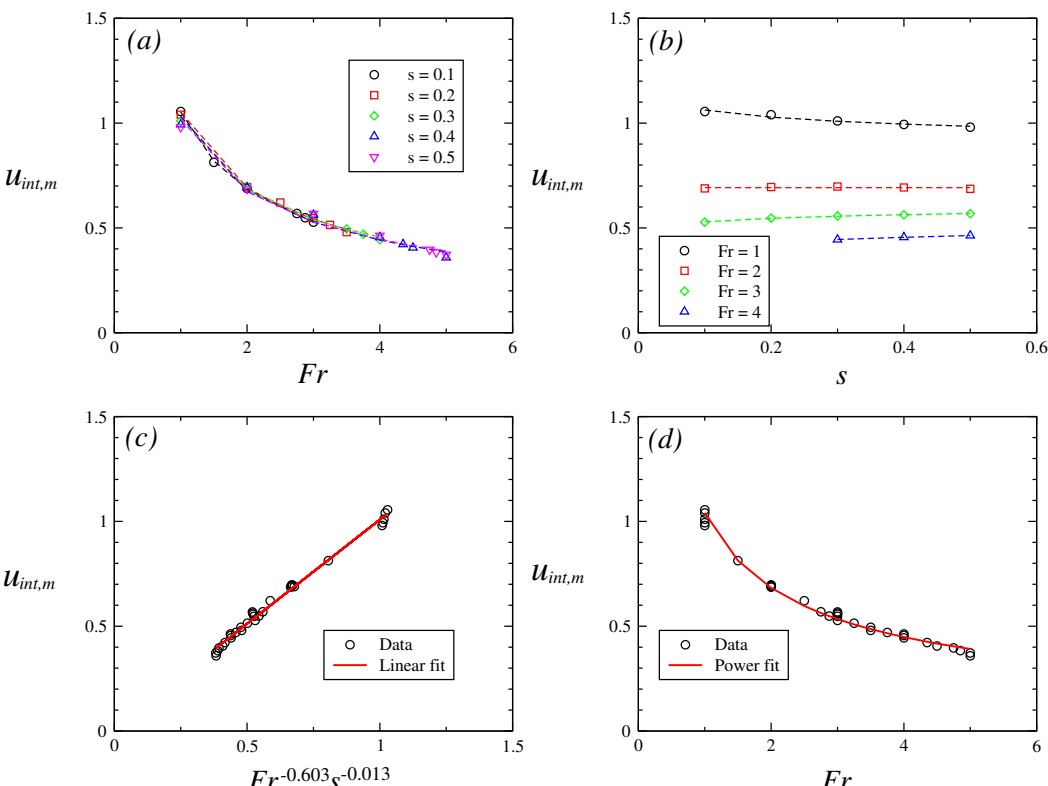

**Figure 14.** (**a**) $u_{int,m}$ plotted against $Fr$ at $s = 0.1, 0.2, 0.3, 0.4$ and $0.5$, (**b**) $u_{int,m}$ plotted against $s$ at $Fr = 1, 2, 3$ and $4$, (**c**) $u_{int,m}$ plotted against $Fr^{-0.603}s^{-0.013}$ over $0.1 \le s \le 0.5$ and $1 \le Fr \le 5$, and (**d**) $u_{int,m}$ plotted against $Fr$ over $0.1 \le s \le 0.5$ and $1 \le Fr \le 5$, respectively.

A similar analysis can also be conducted for $\bar{a}_{u,s1}$, $\bar{a}_{u,s2}$ and $\bar{a}_{u,s3}$, which are the time averages of the moving averages of the acceleration of the intrusion front at Stages 1, 2 and 3, respectively. It is expected that the effects of $Fr$ and $s$ on $\bar{a}_{u,s1}$, $\bar{a}_{u,s2}$ and $\bar{a}_{u,s3}$ should also be quantified by $Fr^a s^b$, although their respective values of $a$ and $b$ are different, as discussed below. It should be noted that the values of $\bar{a}_{u,s1}$ are positive, but the values of $\bar{a}_{u,s2}$ and $\bar{a}_{u,s3}$ are negative. In the subsequent analysis of the results for $\bar{a}_{u,s2}$ and $\bar{a}_{u,s3}$, only their magnitudes are used as their values.

The effects of $Fr$ and $s$ on $\bar{a}_{u,s1}$ are demonstrated by the numerical results presented in Figure 16. As shown in Figure 16a, for each $s$ value, $\bar{a}_{u,s1}$ reduces significantly when $Fr$ increases, which can be quantified by a power-law correlation. However, the effect of $s$ on $\bar{a}_{u,s1}$, similar to that on $\bar{u}_{int,m}$, is negligible, except at $Fr = 1$, as all data with different $s$ values fall approximately on the same power-law curve. This is also clearly demonstrated in Figure 16b which shows that for each $Fr$ value, with the exception of $Fr = 1$, $\bar{a}_{u,s1}$ varies only slightly.

Similarly, the overall effects of $Fr$ and $s$ on $\bar{a}_{u,s1}$ can also be quantified by $Fr^a s^b$, as mentioned above. A multi-variable regression analysis of the results for $\bar{a}_{u,s1}$ from all fountains considered shows that $Fr^{-1.291}s^{-0.024}$ quantifies the combined effects of $Fr$ and $s$ on $\bar{a}_{u,s1}$ reasonably well, and the following correlation is obtained when all fountains are included,

$$\bar{a}_{u,s1} = 0.03154 Fr^{-1.291} s^{-0.024} + 0.00197, \tag{22}$$

with $R^2 = 0.9176$. From this correlation, it is seen that $b = -0.024$, indicating that the effect of $s$ is negligible, which is in agreement with the results presented in Figure 16a,b. Hence, it is expected that the omission of the effect of $s$ on $\bar{a}_{u,s1}$ should not lead to a noticeable

change of $a$. This is verified by the result presented in Figure 16c, which shows that the following power-law correlation can be obtained,

$$\bar{a}_{u,s1} = 0.0385 Fr^{-1.299}, \tag{23}$$

with $R^2 = 0.9362$, where $a = -1.299$, which is almost the same as $a = -1.291$ as obtained in the correlation (22) when the effect of $s$ is included.

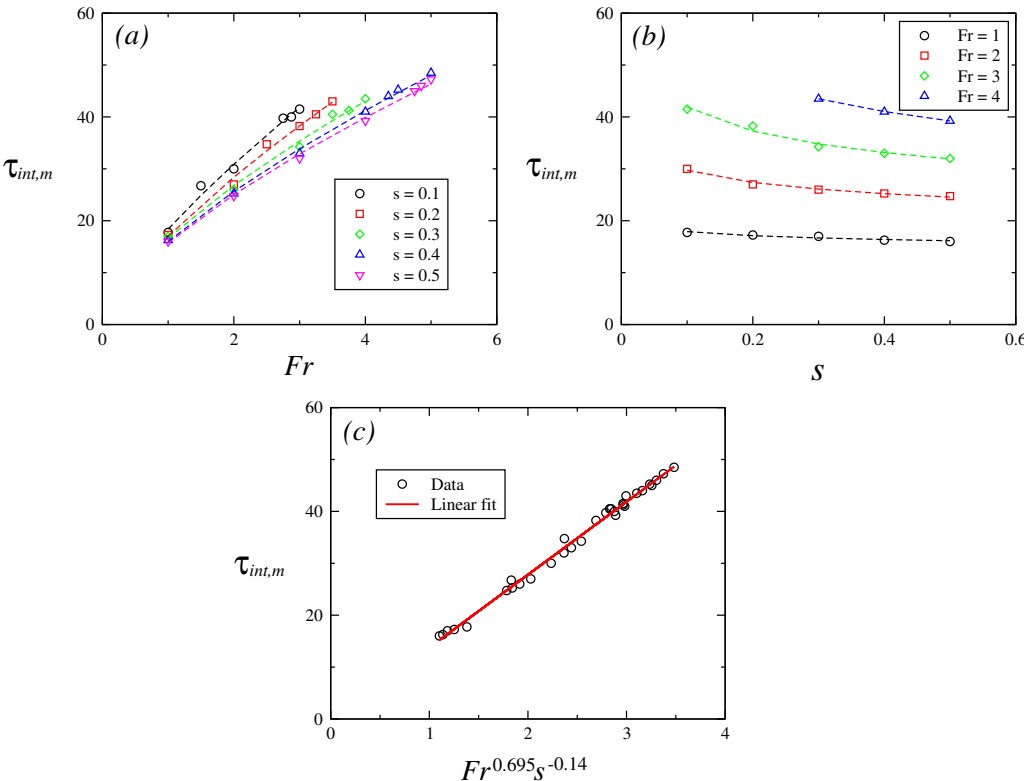

**Figure 15.** (a) $\tau_{int,m}$ plotted against $Fr$ at $s = 0.1, 0.2, 0.3, 0.4$ and $0.5$, (b) $\tau_{int,m}$ plotted against $s$ at $Fr = 1, 2, 3$ and $4$, and (c) $\tau_{int,m}$ plotted against $Fr^{0.695} s^{-0.14}$ over $0.1 \leq s \leq 0.5$ and $1 \leq Fr \leq 5$, respectively.

It should be noted that the relatively large deviations of some data, particularly those at $Fr = 1$, away from the correlations (22) and (23), are due to two reasons. The first one is because for all fountains considered, the moving average interval of 5 (dimensionless time) is used. This leads to the absence of the data within the initial 5 in the determination of the moving average of $u_{int}$, i.e., $\bar{u}_{int}$, in Stage 1, which in turn excludes these data for $\bar{a}_{u,s1}$. As in general, the durations of Stage 1 are relatively small, from about 16 at $Fr = 1$ to the maximum of about 50 at $Fr = 5$ for all fountains considered, the absence of the data within the initial 5 in the determination of the moving average of $u_{int}$ in Stage 1 has a significant effect, particularly at $Fr = 1$ and other small $Fr$ values. The second reason is that it is observed that the changes of $\bar{u}_{int}$ are substantial at different times within Stage 1, particularly at small $Fr$ values, which results in a relatively large inaccuracy of the values obtained for $\bar{a}_{u,s1}$. However, as the changes of $\bar{u}_{int}$ within Stage 2 and Stage 3 are much smaller and additionally, there is no issue with the absence of the data within the initial 5 in the determination of the moving average of $u_{int}$ within these two stages, it is expected that the values of $d$ and $c$ in $Fr^a s^b$ for $\bar{a}_{u,s2}$ and $\bar{a}_{u,s3}$, as well as the relevant correlations are more accurate, as shown below.

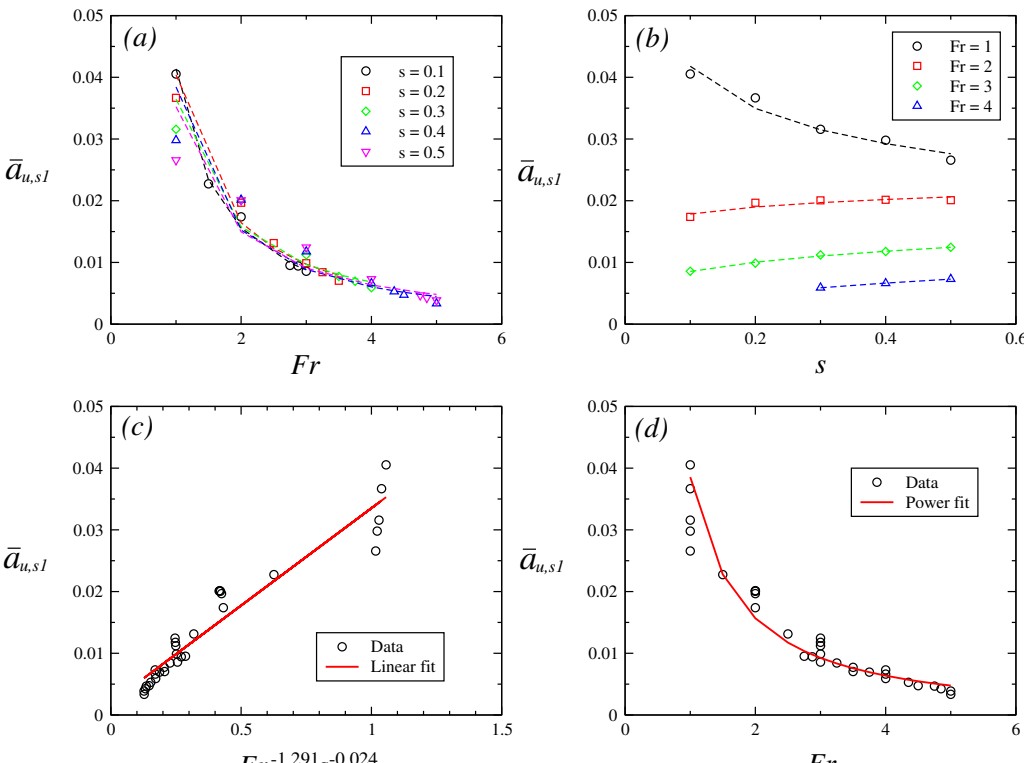

**Figure 16.** (a) $\bar{a}_{u,s1}$ plotted against $Fr$ at $s = 0.1, 0.2, 0.3, 0.4$ and $0.5$, (b) $\bar{a}_{u,s1}$ plotted against $s$ at $Fr = 1, 2, 3$ and $4$, (c) $\bar{a}_{u,s1}$ plotted against $Fr^{-1.291}s^{-0.024}$ over $0.1 \leq s \leq 0.5$ and $1 \leq Fr \leq 5$, and (d) $\bar{a}_{u,s1}$ plotted against $Fr$ over $0.1 \leq s \leq 0.5$ and $1 \leq Fr \leq 5$, respectively.

The effects of $Fr$ and $s$ on $\bar{a}_{u,s2}$ are demonstrated by the numerical results presented in Figure 17. As shown in Figure 17a, for each $s$ value, $\bar{a}_{u,s2}$ reduces significantly when $Fr$ increases, which can be quantified by a power-law correlation. It is also seen that $s$ affects $\bar{a}_{u,s2}$, with its increase leading to a larger $\bar{a}_{u,s2}$ value for each $Fr$ value, as demonstrated in Figure 17b, but the effect of $s$ on $\bar{a}_{u,s2}$ is significant when $Fr = 1$, and it is much smaller for higher $Fr$ values. A multi-variable regression analysis of the results for $\bar{a}_{u,s2}$ from all fountains considered shows that $Fr^{-2.074}s^{0.615}$ quantifies the overall effects of $Fr$ and $s$ on $\bar{a}_{u,s2}$ well, and the following correlation is obtained from the numerical results of all fountains,

$$\bar{a}_{u,s2} = 0.04416 Fr^{-2.074}s^{0.615} + 0.000021, \tag{24}$$

with $R^2 = 0.9742$.

The effects of $Fr$ and $s$ on $\bar{a}_{u,s3}$ are demonstrated by the numerical results presented in Figure 18. As shown in Figure 18a, for each $s$ value, $\bar{a}_{u,s3}$ reduces significantly when $Fr$ increases, which can be quantified by a power-law correlation. It is also seen that $s$ affects $\bar{a}_{u,s2}$, with its increase leading to a larger $\bar{a}_{u,s3}$ value for each $Fr$ value, as demonstrated in Figure 18b, and similar to that for $\bar{a}_{u,s2}$, the effect of $s$ on $\bar{a}_{u,s3}$ is larger when $Fr$ is smaller. A multi-variable regression analysis of the results for $\bar{a}_{u,s3}$ from all fountains considered shows that $Fr^{-0.683}s^{0.35}$ quantifies the overall effects of $Fr$ and $s$ on $\bar{a}_{u,s3}$ relatively well, and the following correlation is obtained when all fountains are included,

$$\bar{a}_{u,s3} = 0.000783 Fr^{-0.683}s^{0.35} - 0.000001, \tag{25}$$

with $R^2 = 0.9482$.

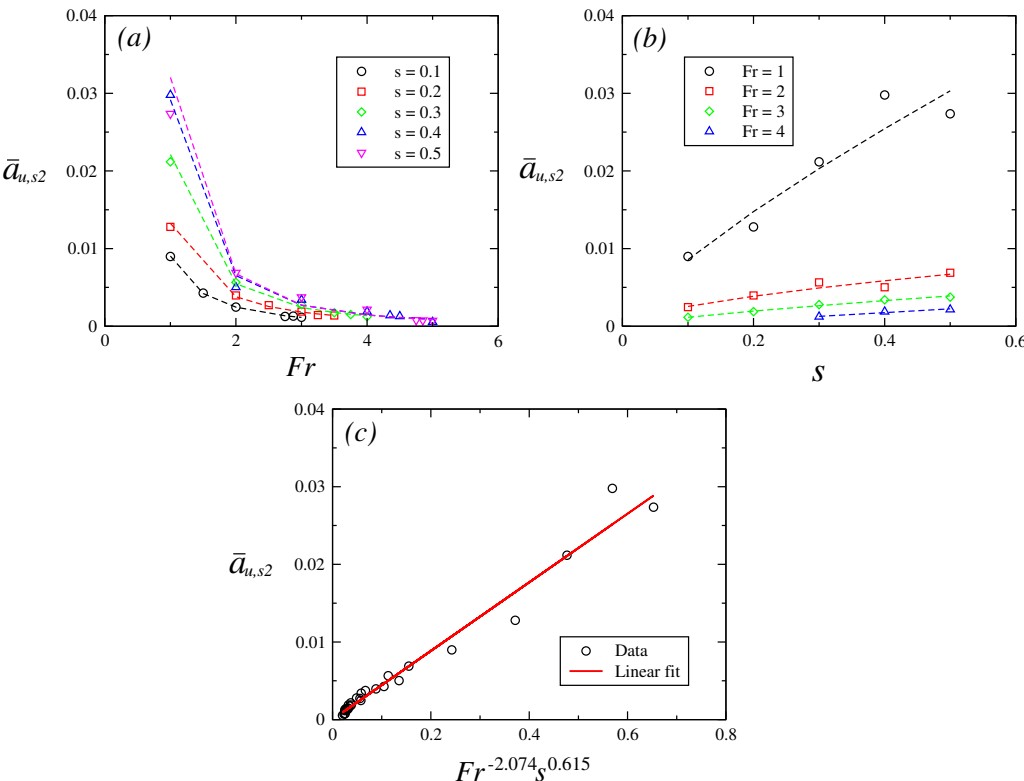

**Figure 17.** (**a**) $\bar{a}_{u,s2}$ plotted against $Fr$ at $s = 0.1, 0.2, 0.3, 0.4$ and $0.5$, (**b**) $\bar{a}_{u,s2}$ plotted against $s$ at $Fr = 1, 2, 3$ and $4$, and (**c**) $\bar{a}_{u,s2}$ plotted against $Fr^{-2.074}s^{0.615}$ over $0.1 \leq s \leq 0.5$ and $1 \leq Fr \leq 5$, respectively.

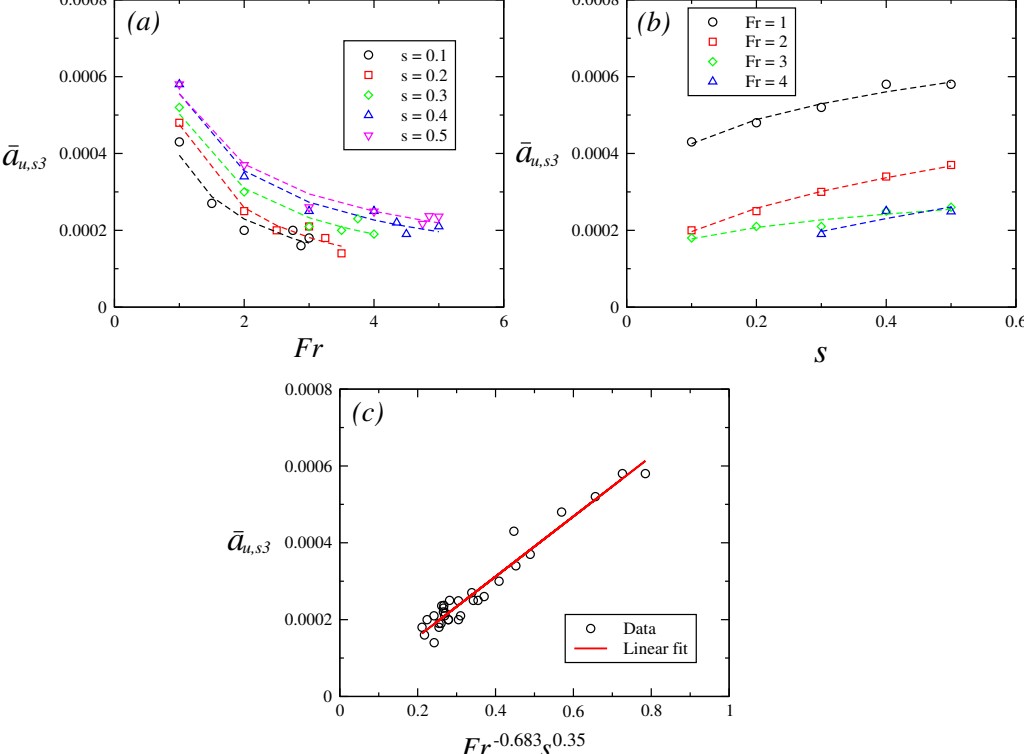

**Figure 18.** (**a**) $\bar{a}_{u,s3}$ plotted against $Fr$ at $s = 0.1, 0.2, 0.3, 0.4$ and $0.5$, (**b**) $\bar{a}_{u,s3}$ plotted against $s$ at $Fr = 1, 2, 3$ and $4$, and (**c**) $\bar{a}_{u,s3}$ plotted against $Fr^{-0.683}s^{0.35}$ over $0.1 \leq s \leq 0.5$ and $1 \leq Fr \leq 5$, respectively.

## 5. Conclusions

The characterization of weak plane fountains in stratified fluids with small $Fr$ and $Re$ values, in which only the symmetric behavior is present, is currently rarely understood. Furthermore, the intrusion is an integral part of a weak symmetric plane fountain, and its behavior is important due to its significant effects on the maximum fountain penetration height. Nonetheless, intrusion in such weak symmetric plane fountains, particularly in stratified ambient fluids, is rarely investigated. A numerical study was thus conducted on the weak symmetric plane fountains in linearly stratified fluids with simulations over $0.1 \leq s \leq 0.5$ and $1 \leq Fr \leq 5$, all at constant $Re = 100$. The parameters selected to investigate the fountain behavior are $z_m$, both initial and time-averaged, the time to attain the initial $z_m$, along with as the velocities of intrusion at different stages.

There are two major differences in behavior between a weak SPF in a homogeneous fluid and that in an LSF. One difference is that the stratification of the ambient fluid stabilizes the symmetry of the weak fountain, which makes the fountain become asymmetric at a larger $Fr$ value. The other difference is that $z_m$ of the SPF in the LSF continues to increase at the FDS, whereas that in the homogeneous fluid is essentially constant. The observed continuous increase of the MFPH in the SPF is largely caused by the intrusion which continuously diminishes with time at the FDS, leading to reduced buoyant force experienced by the fountain fluid. This is especially notable for large $s$ and small $Fr$ values with the MFPH considerably smaller.

The results show that $z_m$ and the associated time are under the effects of $Fr$ and $s$, with the effect of $Fr$ usually stronger than that of $s$. The overall effects of $Fr$ and $s$ can be quantified by $Fr^a s^b$, with the values of $a$ and $b$ varying for different parameters related to $z_m$. With the numerical results obtained for all weak SPFs, empirical correlations are produced in terms of $Fr^a s^b$ for each relevant parameter, which generally predict the results very well. It is further found that if the local Froude number $Fr(z_{m,i})$, which incorporates the effect of $s$, is used instead of $Fr$, the scaling relations using $Fr(z_{m,i})$ only developed by Lin and Armfield [6] for weak SPFs in LSFs basically also work well for the SPFs considered in the present study.

The evolution of the intrusion experiences three distinct stages; Stage 1 is the initial stage, from the formation of the intrusion until $u_{int}$ attains the maximum, in which $u_{int}$ increases continually; this is followed by Stage 2 in which $u_{int}$ reduces monotonically for a period of time; and eventually the intrusion is in Stage 3, which is at the FDS in which $u_{int}$ continually decreases but at rates that are much smaller than those in Stage 2. The results show that both $Fr$ and $s$ have effects on $u_{int}$ and the associated rates of changes with time (accelerations) at different stages and similarly the overall effects of $Fr$ and $s$ on these parameters can also be quantified in terms of $Fr^a s^b$, with different values of of $a$ and $b$. Empirical correlations are obtained in terms of $Fr^a s^b$ for each relevant parameter, which generally predict the results well.

**Supplementary Materials:** The following supporting information can be downloaded at: https://www.mdpi.com/article/10.3390/fluids8040127/s1.

**Author Contributions:** Conceptualization, W.L., M.I.I. and S.W.A.; methodology, M.I.I. and W.L.; software, M.I.I.; validation, M.I.I.; formal analysis, M.I.I., W.L. and M.K.; investigation, M.I.I. and W.L.; resources, W.L.; data curation, M.I.I.; writing—original draft preparation, M.I.I.; writing—review and editing, M.I.I., W.L., S.W.A. and M.K.; visualization, M.I.I.; supervision, W.L. and S.W.A.; project administration, W.L. and S.W.A.; funding acquisition, S.W.A. and W.L. All authors have read and agreed to the published version of the manuscript.

**Funding:** This research received no external funding.

**Data Availability Statement:** The data that support the findings of this study are available from the corresponding author upon reasonable request.

**Conflicts of Interest:** The authors declare no conflict of interest.

## Abbreviations

The following abbreviations are used in this manuscript:

| | |
|---|---|
| FDS | Fully developed stage |
| LSF | Linearly stratified fluid |
| MFPH | Maximum fountain penetration height |
| PF | Plane fountain |
| SPF | Symmetric plane fountain |

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
