# Peer review of "Maximum Penetration Height and Intrusion Speed of Weak Symmetric Plane Fountains in Linearly Stratified Fluids"

_fluids, doi:10.3390/fluids8040127_

Round 1

Reviewer 1 Report

There are some comments for authors to improve the manuscript. Authors must make a major revision before publication.

1.     Clarify the novelty of the work.

2.     Expand the introduction. Authors used for example, [1,2,27,29,30] and [12,13,24,26,31,32], without discussing these works.

3.     All the equations that come from the other publications should be referred to the relevant references.

4.     Information about meshing is missing, also, there should be a study for mesh independence.

5.     There should be a validation study.

6.     What are the shortcomings of the present study that could be considered in future work for researchers?

Reviewer 2 Report

The report presents the outcome of a study on penetration and intrusion of weak symmetric plane fountains. The article was examined based on its applicability, journal's scope, originality, uniqueness, comparison with state of the art, contribution to the body of literature already in existence, methodology, and write-up presentation. The report's contribution to the body of knowledge is significant and novel. Some of the observations are listed below.

Q1. The title should be updated with the fact that the report presents Eq. 18, Eq. 19, Eq. 20, Eq. 21, Eq. 23, Eq. 24, and Eq. 25. Ensure that you use the most suitable words

Q2. The present structure of the abstract seems good but not better and not the best. But, the author may watch the attached videos to gain better insight

Q3. Line 15 - 17, it was written, "Fountains are widely encountered in many natural settings and practical applications, making them a topic that has drawn a remarkable research interest, as reviewed by, e.g., [1–5], and continues to be extensively studied (see, e.g., [6–19]). 1".  This is not true, it is very erroneous to use 19 published facts to present the significance of the study using just a sentence.

Q4. In addition to Q3, the entire structure of the introduction seems not to be good enough.  There is need to restructure the entire introduction section, such definition of the concept, theoretical review, and empirical review could be easily fetched. Note that each paragraph should announce at least an important concept of the study. The authors must describe the rationale for undertaking the study and explain the theoretical framework that the study is based on. You need to provide a background of the problem or issue your research aims to understand or resolve, citing studies to support your arguments. There is a need to restructure the entire introduction section, such definition of the concept, theoretical review, and empirical review could be easily fetched. Note that each paragraph should announce at least an important concept of the study. Recall that the major components of a standard paragraph under the introduction are (a) The definition of the term/concept/idea is needed to understand the exact contribution of the report to the body of knowledge. (b) The theoretical review tells us about the published aim and significant theory. (c) The empirical review tells us about published results within the scope of the subject matter. In view of this, there is need to revise the entire introduction section such that

a) Paragraph 1 should strictly focus on weak plane fountain

b) Paragraph 2 should strictly focus on Froude number

c) Paragraph 3 should strictly focus on

d) Paragraph 4 should strictly focus on

Q4. The report seems to present some expressions which was not introduced properly. First, ensure that the introduction or discussion of results is updated with rationale that leads to Eq. 18, Eq. 19, Eq. 20, Eq. 21, Eq. 23, Eq. 24, and Eq. 25. Is it possible to use a paragraph to announce the significance of Eq. 18, Eq. 19, Eq. 20, Eq. 21, Eq. 23, Eq. 24, and Eq. 25? This is very important because the report seems to focus on these equations

Q5. Line 375, how did you come up with R2 = 0.9482?

Q6. The sections and subsections seems inappropriate. For instance

Line 87 - 88, it was written

3. Qualitative Observation 87

3.1. Development of temperature field 88

Line 116 - 117, it was written

4. Quantitative analysis 116

4.1. MFPH (zm)

Could you note and adopt the list below?

1. Introduction

2. Research Methodology

3. Analysis and Discussion of Results

4. Conclusion

For each section, I would suggest that the authors should highlight your key findings.

Q7. Line 109, it was written, "To investigate the influence of s, Fig. 3..." What is s? Could you replace s with the exact name?

Q8. The authors seems to struggle between physical sciences and mathematics. For instance, line 65, it was written, "...in which an initially quiescent fluid with a constant stratification parameter value is fully filled."

Q9. Back to line 65, is it possible to have constant stratification? What do you mean by stratification? Graphical illustration is needed in section 2.

Q10. Line 377, it was written, "A numerical study was conducted on the weak SPFs in LSFs with simulations over 377 0.1 ≤ s ≤ 0.5 and 1 ≤ Fr ≤ 5, all at constant Re = 100." The sentence is inadequate to announce the aim and objectives of the study before outline the conclusion of the study. Update and revise. Besides, replace SPFs with Symmetric plane fountain?

Q11. Obviously, the Reynold number presented as Eq. 1 is dimensional? However, that is not a characteristic of the Reynold number. Update section 2 with such a fact.

Q12. What about the validation of results? How do we believe that the obtained results are reliable?

Q13. What about the meshing information since you used ANSYS Fluent 13?

Q14. What turbulent model did you use?

Round 2

Reviewer 2 Report

Scientifically, the article can be accepted but requires some language editing.